# ICON in Climate Limited-area Mode (ICON Release Version 2.6.1): a new regional climate model

Trang Van Pham[1], Christian Steger[1], Burkhardt Rockel[2], Klaus Keuler[3], Ingo Kirchner[4], Mariano Mertens[5], Daniel Rieger[1], Günther Zängl[1], and Barbara Früh[1]

[1]Deutscher Wetterdienst, Frankfurterstr. 135, 63067 Offenbach am Main, Germany
[2]Helmholtz-Zentrum Geesthacht, Max-Planck-Str. 1, 21502 Geesthacht, Germany
[3]Brandenburg University of Technology, P.O. 10 13 44, 03013 Cottbus, Germany
[4]Freie Universität Berlin, Carl-Heinrich-Becker-Weg 6-10, 12165 Berlin, Germany
[5]Deutsches Zentrum für Luft- und Raumfahrt, Institut für Physik der Atmosphäre, Oberpfaffenhofen, Germany

**Correspondence:** Trang Van Pham (trang.pham-van@dwd.de)

**Abstract.** For the first time the limited-area mode of the new weather and climate model ICON has been used for a continuous long-term regional climate simulation over Europe. Built upon the limited-area mode of ICON (ICON-LAM), ICON-CLM (ICON in Climate Limited-area Mode, hereafter ICON-CLM, available in ICON Release Version 2.6.1) is an adaptation for climate applications. A first version of ICON-CLM is now available and has already been integrated into a starter package (ICON-CLM_SP Version Beta1). The starter package provides users with a technical infrastructure that facilitates long-term simulations as well as model evaluation and test routines. ICON-CLM and ICON-CLM_SP were successfully installed and tested on two different computing systems. Tests with different domain decompositions showed bit-identical results, and no systematic outstanding differences were found in the results with different model time steps. ICON-CLM was also able to reproduce the large-scale atmospheric information from the global driving model. Comparison was done between ICON-CLM and COSMO-CLM (the recommended model configuration by the CLM-Community) performance. For that, an evaluation run of ICON-CLM with ERA-Interim boundary conditions was carried out with the set-up similar to the COSMO-CLM recommended optimal set-up. ICON-CLM results showed biases in the same range as those of COSMO-CLM for all evaluated surface variables. While this COSMO-CLM simulation was carried out with the latest model version which has been developed and was carefully tuned for climate simulations on the European domain, ICON-CLM was not tuned yet. Nevertheless, ICON-CLM showed a better performance for air temperature, its daily extremes, and slightly better for total cloud cover. For precipitation and mean sea level pressure, COSMO-CLM were closer to observations than ICON-CLM. However, as ICON-CLM is still in the early stage of development, there is still much room for improvement.

## 1 Background information

In 1999, the limited-area weather forecast model LM (Lokalmodell, Doms and Schättler (1999), later COSMO, Baldauf et al. (2011)), which was developed by the Deutscher Wetterdienst (DWD, the German Meteorological Service), went operational together with the global model GME (Majewski and Ritter, 2002). A few years later, it was renamed into "COSMO model" in

order to reflect that further development has become a joint task of the COnsortium for Small scale MOdelling (COSMO). In 2002, the Climate Limited-area Modeling Community (CLM-Community) developed the first version of the regional climate model named CLM. In 2007, the developments in COSMO and CLM were recombined and a first unified version of the weather forecast and climate modes, named COSMO-CLM (Rockel et al., 2008), was released.

In 2001, a cooperation between DWD and Max-Planck Institute for Meteorology (MPI-M) was initiated, with the aim to develop a new modelling system for weather forecast and climate prediction. The new system was intended to replace the existing system COSMO/GME for operational weather forecast on one side and, on the other side, the global climate and earth system model ECHAM6/MPI-ESM (Stevens et al., 2013; Giorgetta et al., 2013). As a result of this initiative, the global numerical weather forecast model ICON (Icosahedral Nonhydrostatic) (Zängl et al., 2015) was developed and replaced GME

as the operational model at DWD on the 20th of January 2015. As a next step, in December 2016, ICON-EU-Nest, the regional ICON on the European domain interactively nested within the global ICON, replaced COSMO-EU (high resolution COSMO model configuration for Europe) for higher-resolution forecasts on the European domain. In the second half of 2020, the convection permitting configuration of ICON-LAM (ICON-D2) became pre-operational. According to the plans, ICON-D2 will replace the high resolution COSMO-D2 for the German domain early 2021 and DWD will stop the operational use of the

COSMO model after more than 20 years. This implies that the next unification of COSMO and COSMO-CLM (COSMO 6), scheduled for the end of 2020, will be the last one. Afterwards, the support for COSMO and COSMO-CLM will be gradually reduced.

    In this work, we prepared state-of-the-art tools for climate applications for the upcoming years. Starting in 2017, DWD and the CLM-Community decided to develop a new regional climate model (ICON-CLM) based on the Limited-Area Mode of

ICON (ICON-LAM). The preparation of ICON-CLM was triggered at DWD in the project ProWaS (Projection Service for Waterways and Shipping) – a joint pilot program of several German Federal Agencies – to prepare a regular federal forecasting and projection service about the influence of climate change on coastal and waterway traffic.

    ICON can be used on a wide range of scales from climate projection, climate prediction, numerical weather prediction (NWP) down to large-eddy simulations (Heinze et al., 2017). For these different scales, there is a number of different modes

as shown in Figure 1. Generally, there are three different physics packages available: the NWP, the ECHAM physics, and the large-eddy physics (LES physics). Within the first physics package, at global scale, the ICON-NWP is used for operational weather forecasting at DWD. ICON-EU-Nest is the regional ICON on the European domain nested within the global ICON-NWP. ICON-LAM denotes the limited-area mode of ICON-NWP, which currently is available for the NWP and large-eddy configurations. ICON-LEM (ICON-Large Eddy Model) applies the physics package dedicated to large eddy simulations to

study processes such as cloud, convection and turbulence on a very high resolution grid. Within the ECHAM physics package, the global atmospheric model ICON-A (Giorgetta et al., 2018), originating from the general circulation model ECHAM6, is used for global climate simulations. This configuration is coupled to the global ocean model ICON-O (Korn, 2017) and the land and biosphere model JSBACH (Brovkin et al., 2013) within the ICON Earth System Model (ICON-ESM). The feature for one- or two-way nested sub-domains with grid refinement available in the NWP physics package (ICON-EU-Nest) has

recently also been transferred to the ECHAM package by DWD (ICON-EUClim).

ICON-CLM builds upon ICON-LAM, and currently contains a set of technical adaptations for climate applications.

The aim of this paper is to introduce the new regional climate mode of ICON, ICON-CLM, along with its starter package ICON-CLM_SP, a supporting infrastructure needed to perform long-term simulations. Different technical tests and tests on the impact of prescribing upper boundary conditions interpolated from re-analysis data were carried out. A long evaluation simulation driven by ERA-Interim re-analysis (Dee et al., 2011) was conducted over a period of 20 years and the results were compared to the evaluation simulation of the latest COSMO-CLM version recommended by the CLM-Community (called "recommended version", CCLM 5.0 clm9). The paper is structured as follows: Some related general information on ICON-NWP and ICON-LAM is provided in Section 2. The adaptations in model source code and technical infrastructure are described in Section 3. Section 4 gives details of the ICON-CLM model configuration and setup for the evaluation run as well as the evaluation methods we used. Results of the technical tests and of the evaluation run in comparison to observational data and to the results of the latest COSMO-CLM version are shown in Section 5. Conclusions are provided in Section 6.

## 2   General information on ICON-NWP and ICON-LAM

All ICON models in the ICON family (Figure 1) utilize unstructured triangular grids that originate from a spherical icosahedron with 20 equal sized triangles. ICON horizontal grid is denoted as $R_nB_k$; this is a triangular grid generated from the original icosahedron by first dividing the edges into $n$ parts, followed by $k$ subsequent edge bisections. The division of the edges into n equal parts gives $n^2$ spherical triangles within the original triangle. In the second step, each triangle is again subdivided into 4 smaller triangles. The resulting grid $R_nB_k$ has, therefore, the total number of triangle cells $n_{cells}$ and the number of edges $n_{edges}$ calculated from the following formulas:

$$n_{cells} = 20n^2 4^k; \quad n_{edges} = 30n^2 4^k \tag{1}$$

The average area of the triangles is then equal to the earth surface divided by the number of triangles $n_{cells}$. The effective grid is defined as a square which has the same area as the average triangle area, and the effective grid size $\overline{\Delta x}$ is defined as:

$$\overline{\Delta x} = \sqrt{S_{earth}/n_{cells}} = \sqrt{\frac{4R_{earth}^2\pi}{n_{cells}}} = \frac{R_{earth}}{n2^k}\sqrt{\frac{\pi}{5}} \approx 5050/(n2^k) \quad [km] \tag{2}$$

It can be drawn from Eq. 2 that the effective grid size is around 0.658 size of the average triangle grid size. Some characteristics of the model grids used and mentioned in this work are listed in Table 1. Figure 2b visualizes the R2B8 grid extracted from the EURO-CORDEX domain (marked in red box from Figure 2a) .

The vertical layer distribution in the ICON models is a height-based coordinate system following the terrain, with denser layers near the earth surface and gradually changing to constant height model levels above a certain height. Two options for the height-based terrain-following vertical coordinate are offered in the ICON models, the terrain-following hybrid Gal-Chen coordinate (Simmons and Burridge, 1981) and the Smooth Level Vertical SLEVE coordinate (Schär et al., 2002; Leuenberger et al., 2010). With SLEVE (used in the simulations in this paper), the influence of small-scale terrain features decays more quickly with height than the large-scale features in order to obtain smooth vertical coordinate levels at mid and upper levels.

The vertical coordinate is a function of model top height, the layer thickness of the lowermost layer, the total number of vertical layers, and the stretch factor which controls the distribution of the model levels. Users can define the vertical model levels by setting these controlling parameters via the model namelist.

Outputs in the ICON models can be written out in GRIB or Net-CDF format. Options for outputting on the ICON native grid or regular lat-lon grid or rotated lat-lon grid are available. Outputs can be written with individual or multiple fields in an output file. Users can define how many output steps in one output file and the output frequency. It is possible to have outputs with different intervals, for example hourly precipitation, daily temperature and monthly mean mean sea level pressure (MSLP) in the same run.

At the lateral boundary of the limited area domain, a sponge layer is applied, within which the internal flow is gradually relaxed towards the external boundary data. At the outer most area of the limited domain, the "lateral boundary zone" is a stripe fixed with 4 cell rows. Here the external boundary data are simply prescribed. After the outer 4 cell rows is the "lateral boundary nudging zone". The width of this nudging zone can be defined in the namelist setting of ICON with the minimum value of 8 cell rows to prevent the boundary artifacts. The nudging coefficient gradually reduces from the outer to the inner edge of the "lateral boundary nudging zone", making the influence of the prescribed external data weaker. The strength of the nudging can be controlled by the maximum relaxation coefficient in the model namelist.

For the upper boundary, ICON-LAM offers an option of prescribing the upper boundary conditions by using the same driving data source as for the lateral boundary conditions (nudging option). Users can define the height of the nudging zone as well as the nudging coefficients for the horizontal wind and for the thermodynamic variables via namelist settings. If this vertical nudging option is turned off, a Rayleigh damping is applied to the vertical wind speed within the damping layer in order to prevent unphysical reflection of vertically propagating gravity waves.

## 3   Model development

As the Limited-Area Mode of ICON, which ICON-CLM builds upon, has originally been developed for NWP applications, several adaptations and technical extensions were necessary to prepare the model for climate applications. Apart from the adjustments in the code, long-term climate simulations require a technical infrastructure for data and job management. Such an infrastructure has also been developed based on the existing infrastructure of COSMO-CLM.

### 3.1   The regional climate model ICON-CLM

Weather forecasting, which predicts the state of the atmosphere only up to about 2 weeks in advance, often does not involve the development of the ocean state. The ocean surface condition, hence, is often kept constant during the forecasts in weather prediction models or just slightly adjusted with a climatological trend for the forecast period. Thus, in ICON-LAM there is only option to update the sea surface temperature (SST) and sea-ice cover monthly. For ICON-CLM, we want to have a flexible option to update of SST and sea ice from external data at a user-defined interval. For this purpose, an option for flexible update frequencies of these boundary conditions was implemented in ICON-CLM. Time-dependent SST and sea-ice data can now be

read from external data files and are fed to ICON-CLM with shorter intervals than one month (e.g. 1 hourly or 6 hourly). The user can select this option of frequent update of SST and sea ice via namelist settings. The external SST and sea ice data must be prepared and remapped to the ICON grid.

Similarly, the greenhouse gas (GHG) values are usually kept constant in weather forecast models, because the changes
during the forecast period are negligible. In climate projections, however, it is necessary to use the time-dependent GHGs as provided by corresponding GHG scenarios. Such an option was already available in ICON, but only in combination with the ECHAM physics package. The corresponding read routine was therefore extended so that it works for the NWP physics as well. Some additions to the NWP radiation scheme were made with respect to the GHG vertical profile with a new option to retrieve the profile from external gas data. A file that contains yearly values of $CO_2$, $CH_4$, $N_2O$ and Chlorofluorocarbons (CFC)
for all years of the experiment needs to be prepared in advance. These features of the time-dependent SST and GHG were largely based on the corresponding implementations in the ICON-A (Giorgetta et al., 2018).

In the NWP configuration of ICON-LAM, the number of soil layers is always constant with eight layers. The depths of half soil layers are also fixed at values between 5 mm and 14.5 m. However, for climate simulations in domains other than Europe (e.g. Africa, Asia) or to achieve better simulation of the soil variables for the European domain, it is usually reasonable to
15 adjust these soil parameters. Therefore, an option for a flexible number and depth of the soil layers has been implemented in the ICON-CLM code.

The input/output of ICON-CLM has also been adjusted to have more flexibility. In NWP mode, the precipitation data are accumulated from the start of the forecast till the end without any reset. This is suitable for short weather forecasts, but for long climate simulations, this procedure is inconvenient and could, in the worst case, cause problems due to data imprecision.
Furthermore, the maximum and minimum 2-m temperature values are calculated for 6 hourly intervals in NWP applications, while for climate simulations the standard for these output variables is usually 24 hours. To control this flexibility extension, new namelist parameters were introduced in ICON-CLM.

At the lateral boundaries, ICON-LAM requires, by default, information on cloud liquid water content and cloud ice water content from the global forcing data. These input fields are usually available if the ICON-CLM lateral boundary conditions are
25 taken from reanalysis data like ERA-Interim. But if global climate projections are used as lateral boundary conditions, these fields are usually not provided. Thus, the model code has been adjusted so that if cloud liquid water content and cloud ice content are not available in the lateral boundary data, these variables are initialized with zero.

Table 2 provides an overview of some differences between COSMO-CLM and ICON-CLM.

## 3.2 The starter package ICON-CLM_SP

In order to facilitate long-term climate simulations, we developed a run time infrastructure called starter package and a separate evaluation tool. Both are provided along with the ICON-CLM model source code. The starter package ICON-CLM_SP contains a run routine, a climatological testsuite, all necessary utilities and configure scripts for different super computing environments. At the moment, two system settings for Nec-Aurora (DWD) and Atos/Bull (DKRZ) are supported and tested (note that our

ICON-CLM simulations in this paper were done on the DWD Cray XC40, this computer was replaced by the Nec-Aurora afterwards). Settings for other machines could be easily added if necessary.

The run routine in ICON-CLM_SP, called "subchain", was adapted from the routine of the existing COSMO-CLM package. The "subchain" contains five sub-routines for input preparation (prep), converting input data (conv2icon), ICON-CLM job management (icon), archiving (arch) and postprocessing (post) of the model output. Sub-routine "prep" copies and checks all the global forcing data as input for "conv2icon". Then "conv2icon" preprocesses and interpolates the initial data and the lateral, lower and upper boundary data onto the ICON-CLM model grid for the current model simulation. Sub-routine "icon" does the job management for ICON-CLM model. After that, all model output data are compressed by "arch" and some post-processing steps like the provision of time series of selected output variables are done in "post". Usually the simulations in ICON-CLM are done per calendar month with restarts at the end of each month, however the simulation duration can be easily customized by changing in the sub-routine "icon".

A climatological test suite (CTS) was also created based on the CTS from COSMO-CLM. In the CTS, 5-year test simulations can be done automatically with "subchain". The users can choose one simulation as a reference. The test simulations then will be compared with the reference simulation with respect to observational data (E-OBS and CRU, see Section 4.2 for more details) by an extra sub-routine called "eval". At the end, the results are visualized with standardized plots. This CTS was built for the purpose of testing different versions of model source codes, or different setups of the same model version. Hence, it is a very helpful tool for model development and tuning.

Besides the sub-routine "eval" in CTS, a separate evaluation tool called "ETOOLS" was also adapted from the COSMO-CLM evaluation tool. This tool provides comparisons of the simulation results with observation data sets and creates standardized plots to visualize the results. In order to facilitate the transition from COSMO-CLM to ICON-CLM for the users, both ICON-CLM_SP and ETOOLS were created such that the "look and feel" as well as the usage of the software packages is as similar as possible to the corresponding packages that exist for the COSMO-CLM model. The output structure of ICON-CLM or post-processed time series from "subchain/post" are also similar to those of COSMO-CLM for the same reason. On this account, users should be able to use all existing scripts and programs that were developed for COSMO-CLM output also for ICON-CLM data.

## 4 Data and methods

### 4.1 Model configuration and experiment setup

All of the ICON-CLM tests described afterwards and the evaluation run were performed at the resolution R2B8 (approximately 10 km) on a domain (Figure 2a) completely covering the EURO-CORDEX domain (Giorgi et al., 2009). The model atmosphere is divided into 70 vertical layers and the model top is at a height of 30 km or 10 hPa. The soil in ICON-CLM contains eight layers down to a depth of 14.5 m. ICON-CLM was driven at the lateral and lower boundaries by ERA-Interim at 6 hourly intervals. The model atmosphere was initialized with ERA-Interim data. The soil temperature and soil moisture were taken from a previous simulation which is long enough to ensure that the spin-up of the soil has been completed. The monthly Tegen

aerosol climatology (Tegen et al., 1997) and the ozone climatology from Global Earth system Monitoring using Satellite and in-situ data (GEMS) were used in our simulations. For the upper boundary, as described in Section 2, there are two options: (1) Using the driving data and nudging gradually in the relaxation zone; (2) Damping the vertical wind beneath the upper boundary. To assess the impact of these two options, two 10-year simulations (1979-1988) were done with the same set-up,

with and without global data nudging. Analysis from these 10-year runs resulted in very minor differences on surface variables and none of the options showed any advantage over the other. For the evaluation run, we chose the option with nudging data at the upper boundary with ERA-Interim data, as later we wanted to compare the results with those from a COSMO-CLM run using a similar nudging option. The nudging zone started from the height of 12 km to the model top of atmosphere (30 km). Variables which are nudged within this layer is the horizontal wind and the thermodynamic variables (air pressure

and temperature) with nudging coefficient 0.04 and 0.075 respectively.

In order to find a suitable model configuration for ICON-CLM at the resolution R2B8, an optimized namelist setup was used, namely the setup from ICON-NWP for R3B7 with nested domain on R3B8 grid (approximately 13 km and 6.5 km respectively). The tuning parameters were taken over from the global settings. In this setting, the Tiedtke/Bechtold (Bechtold et al., 2008) convection parameterization scheme and the Rapid Radiation Transfer Model (RRTM) radiation scheme (Mlawer

et al., 1997) were used. These setups were checked to make sure that they are appropriate for climate applications and were used in all simulations.

Former simulations (not published) with COSMO-CLM showed that in some cases the model results depended on the chosen model time steps. There was one particular time step that led to larger biases in precipitation and surface pressure, especially over the Alps and the south western area of the EURO-CORDEX domain. This issue has been analysed by the

CLM-Community but is still not fully understood yet. To ensure that such a dependency of the results on the model time step is not present in ICON-CLM, different fast physics/advection time step (hereafter: time step) choices were tested. At R2B8 (approximately 10 km) resolution, the time step should not exceed 120 seconds for stability reasons. With the common model and experiment setups described above, we carried out multiple one-year simulations for the year 1979 with time steps of 60, 80, 90, 100 and 120 seconds. Figure 3 shows the biases compared to reference data of 2-m temperature, total precipitation,

MSLP and total cloud cover. The biases were averaged for each month and for the Alpine region (sub-region denoted AL in Figure 2c). Colors show the different time step experiments. The biases for all variables from any particular time-step experiment were small and did not stand out from the rest. Similar results were found for all other sub-regions and therefore are not shown here. The annual and seasonal biases of these multiple one-year simulations were also very similar with all time steps (not shown here).

Because there is no big difference in the model results depending on the choice of time step, for experiments at spatial resolution R2B8, we chose the time step of 90 seconds due to the computational efficiency and stability, the time step of COSMO-CLM at similar horizontal resolution is 100 seconds. An evaluation run was carried out for the EURO-CORDEX domain at the resolution R2B8. The simulation period is 1979 to 2000. The model and experiment setups were the common setups as described above, this evaluation run is later referred to as ICLM-REF.

The results of ICLM-REF were compared to the reference experiment of the recent recommended version of COSMO-CLM (v5.0_clm9). This COSMO-CLM simulation is later referred to as CCLM-REF. This COSMO-CLM setup showed the best performance for the EURO-CORDEX domain in an inter-comparison with a large number of setups which was performed within the COPAT project (COordinated Parameter Tuning, a project providing a thorough evaluation of a large number of COSMO-CLM configurations to come up with a recommended version) in the CLM-Community. The simulation period is 1979 to 2000; the model resolution is 0.165°, also about 10 km like in ICLM-REF. The initial, lower and lateral boundary data are taken from ERA-Interim. A sponge layer with Rayleigh damping in the upper levels of COSMO-CLM domain was used. The damping was done against the external boundary values, similarly to the nudging at the model top in ICLM-REF simulation and thus the results from both experiments are comparable.

## 4.2 Evaluation methods

For model assessment and evaluation, output fields from six variables were analysed, namely 2-m temperature, daily maximum and minimum values of 2-m temperature, MSLP, total precipitation and total cloud cover. Monthly average values of these variables were calculated and used for further analysis. For total precipitation, the monthly accumulated amounts were calculated. Parts of the evaluation were carried out based on seasonal averages. The following definitions and abbreviations of the seasons are used in this paper: winter - December, January, February (DJF), spring - March, April, May (MAM), summer - June, July, August (JJA), and autumn - September, October, November (SON). Results were averaged for eights sub-regions as already used in the PRUDENCE projects (described by Christensen and Christensen (2007)). These sub-regions are: British Isles (BI), Iberian Peninsula (IP), France (FR), Mid-Europe (ME), Scandinavia (SC), Alps (AL), Mediterranean (MD) and Eastern Europe (EA) (shown in Figure 2c).

For the evaluation of mean values of 2-m temperature, daily maximum and minimum 2-m temperature, MSLP and total precipitation, the E-OBS dataset (Haylock et al., 2008; Van den Besselaar et al., 2011) was used as reference data. E-OBS is a 0.25° gridded daily dataset covering all of Europe. The data are available over land from quite recent back to 1950. In order to compare 2-m temperature data from different datasets (ICLM-REF, CCLM-REF and E-OBS), a height correction was performed. The model temperature values at E-OBS 2-m height were calculated based on the differences between model and E-OBS surface elevation and the moist adiabatic lapse rate (0.0065 K/m). The reference cloud data, which was used for assessment of the model cloud cover, are CRU TS data (Harris et al., 2014). This is a monthly gridded dataset at 0.5° resolution, available globally over land area. The dataset covers the period from 1901 to 2013.

For all ICON-CLM simulations in this paper, the outputs were written in Net-CDF format and on the rotated lat-lon grid as in CCLM-REF. Because this rotated lat-lon grid is finer than E-OBS and CRU grids, these data were remapped to the regular lat-lon grids with the same spatial resolution as the observational data for the purpose of comparison. The E-OBS and CRU datasets contain data only over land, therefore the evaluation in this paper (e.g. areal averaged fields) were done using only land points.

The evaluation within COPAT for 2-m temperature, MSLP and cloud cover was done using E-OBS version 10.0 and CRU version 3.22 as reference data. Therefore, in order to compare our evaluation to the one from COPAT, we also used the same

versions of the data sets. The comparison period is 20 years from 1981 to 2000, the same as the evaluation period in COPAT. The reference total precipitation data was taken from E-OBS version 12.0 because this dataset (among versions from 10.0 to 17.0) shows the fewest missing data for precipitation over the area of Poland.

Some important climate indices (listed in Table 3) were calculated from ICLM-REF, CCLM-REF and E-OBS 2-m temper-
ature and total precipitation data and averaged over the period 1981-2000. The number of days that fulfils the definition (in Table 3) was counted for each horizontal grid cell, then averaged over a sub-region.

Root-mean-square error (RMSE) was calculated from the model and observed monthly values (monthly aggregated values for precipitation and monthly mean values for other variables):

$$RMSE = \sqrt{\frac{1}{N} \sum_{i=1}^{n} (S_i - O_i)^2} \tag{3}$$

where $S_i$ and $O_i$ are the model (ICLM-REF or CCLM-REF) and reference data (E-OBS or CRU TS) monthly values, respectively, averaged over the sub-regions considered at the $i$th month; $N$ is the total number of months in the evaluation period 1981-2000.

To compare the spatial variability of the data, spatial standard deviation (STDEV) was also calculated from time-averaged fields of model and observed data. The STDEV ratio ($STDEV_{model}/STDEV_{observation}$) was calculated for the ease of
comparing the spatial variation of the two model data with respect to the observation. The model with STDEV ratio closer to one better represents the spatial variation of the observation data.

The results of air temperature from ICLM-REF were also compared to those of other regional climate models (RCMs) within the EURO-CORDEX experiments. A thorough evaluation of several EURO-CORDEX ensembles is presented in Kotlarski et al. (2014). One ensemble, called EUR-11, has quite similar set-up to our ICLM-REF set-up. The horizontal resolution of the
RCMs in this EUR-11 ensemble is about 12 km; these simulations were also driven by ERA-Interim. The simulation period, however, differs from ours, 1989-2009 instead of 1981-2000. Nevertheless, this comparison for sure gives us some knowledge about ICON-CLM performance relatively to other state-of-the-art RCMs. In Figure 5 and B1 in Kotlarski et al. (2014)), some similar analysis to ours were done for 9 different RCMs, seasonal means of temperature bias are shown for 8 PRUDENCE sub-regions. These biases were also calculated against E-OBS data like in our evaluation, though an older version of E-OBS
data were used in their paper. These figures were used to compare to our figures from ICLM-REF.

## 5 Testing and evaluating ICON-CLM

### 5.1 Technical tests

After the technical adaptation in the ICON model source code to enable long-term climate simulations, a number of technical tests was performed. First, the influence of different domain decompositions for parallelization on the simulation results was
tested at the super computer Cray XC40 at DWD. The domain of ICON models is split by a built-in geometric subdivision according to the number of processors used for calculation. Tests with ICON-CLM were done using 120, 240, 480, 960

computer processors, each test simulated the year 1979. As these tests are one year long each, they contain multiple monthly restarts of the model. The model outputs were checked for all climate variables evaluated in this paper. Results were binary identical, independent of the numbers of processors used for the simulating. ICON-CLM showed the ability to produce the same results and to restart properly with a variety of domain decompositions.

Repeating tests starting from the same restart state were also carried out. ICON-CLM was restarted at the time point 01.01.1980 at 00:00 UTC multiple times from the same restart file on the Cray XC40. Each simulation was carried out for one month from 01.01.1980 to 31.01.1980. Model outputs were checked for all model calculation steps and for all climate variables evaluated in this paper. In addition, the restart files created at the end of each test for the simulation time 01.02.1980 at 00:00 UTC were also compared. The results from these repeating tests showed binary identical values.

Two additional restart tests were also done. In the first test, ICON-CLM was run for two months from 01.01.1980 to 29.02.1980 without restarting in between. And in the second test, ICON-CLM also simulated these two months but was restarted at 01.02.1980 00:00 UTC. The model outputs were compared, with focus on the period after the restart of the second test. Results showed that restarting did not introduce any difference in model diagnostics.

Short tests up to few months were also done on a different computing system, the Atos/Bull at the German Climate Comput-
15 ing Centre (DKRZ). ICON-CLM showed the ability to run stably on at least two machines. Performance tests were done for ICON-CLM and COSMO-CLM on the Atos/Bull. The two models were run in one-month simulations on European domain. The horizontal resolution is roughly 50 km, same number of computer processors was used. ICON-CLM was about 15% faster than COSMO-CLM in these tests.

## 5.2   Evaluation and comparison with COSMO-CLM

Initially a longer simulation with ICON-CLM (30 years) was planned and carried out. But since the data from CCLM-REF are only available to us in the period 1981-2000, only data in this period were taken for evaluation.

### 5.2.1   Air temperature

The 2-m temperature bias of ICLM-REF and CCLM-REF was mostly within -1.5 K to 1.5 K relative to E-OBS data. Figure 4 shows the mean annual biases against E-OBS data over the 20 year period from 1981 to 2000 for ICLM-REF and CCLM-REF
data for the entire domain. Both experiments had dominant negative bias over Eastern Europe, with ICLM-REF to a lesser extend. ICLM-REF had a warm bias in most parts of Sweden and southern Russia, while CCLM-REF showed a cold bias in these regions. The Balkan region is well known for large air temperature biases in COSMO-CLM simulations (Anders and Rockel, 2009; Pham et al., 2014; Trusilova et al., 2014); which also occurred in ICLM-REF with values up to 1.5 K.

The seasonal temperature bias in Figure 5 shows that the median of the bias in the sub-regions ranged from -0.58 K to
30 0.81 K for ICLM-REF and -1.13 K to 1.20 K for CCLM-REF. While in many of the sub-regions and seasons relatively small bias was found for ICLM-REF (bias median in the order of 0.01 K), CCLM-REF had usually larger biases. Both models had smaller biases in winter and autumn and larger biases in spring and summer. Besides some sub-regions like British Isles, Mid-Europe, France, Iberian Peninsula where small biases were found (especially from ICLM-REF), sub-regions Scandinavia,

Alps, Mediterranean and Eastern Europe showed larger biases. The latter sub-regions also showed larger variability of biases in space, with the bias range from -3.7 K to 3.4 K. CCLM-REF had almost always more variability in temperature bias than ICLM-REF. The differences in spatial variability in CCLM-REF were exceptionally strong in Scandinavia in winter and summer.

Compared with the nine RCMs in EURO-CORDEX ensemble EUR-11 (Kotlarski et al., 2014), ICLM-REF showed similar magnitude of biases for all four seasons. Our ICLM-REF results from Figure 5 were placed in comparison to Figure 5 and B1 in Kotlarski et al. (2014). ICLM-REF and other EURO-CORDEX experiments tended to have negative biases for air temperature in winter. In spring the opposite was observed, ICLM-REF gave positive bias for most of the sub-regions, while EUR-11 continued to have negative biases. In all seasons, ICLM-REF biases stayed well fit within the spread of the EUR-11 ensemble,

and had smaller biases compared with some other RCMs. In winter, for example, it can be seen on the spread of the solid circles (representing EUR-11 ensemble) in Figure 5 and B1 in Kotlarski et al. (2014) that the temperature biases of EUR-11 ensemble were up to -4 K and -3 K for sub-region Alps and Mediterranean respectively; and for other sub-regions up to around -2 K. Meanwhile, ICLM-REF had mean biases quite closed to 0 for most of the sub-regions (Figure 5, DJF).

For daily max/min temperature bias, CCLM-REF and ICLM-REF showed opposite results. CCLM-REF tended to under-

estimate the maximum 2-m temperature, while ICLM-REF overestimated the values. This can be most clearly seen in winter and autumn (Figure 6), where the medians of almost all ICLM-REF biases were positive and of most CCLM-REF biases were negative. ICLM-REF clearly had smaller median biases ranging from -0.02 to 2.34 K, while the range in CCLM-REF was -2.5 to 1.2 K. In summer, the bias was larger for both models. CCLM-REF had larger spatial differences among the sub-regions than ICLM-REF. Similarly, the bias of the minimum 2-m temperature of ICLM-REF was reduced compared to CCLM-REF (-1.2

K to 0.5K compared to -0.6 K and 1.8K, Figure 7). ICLM-REF slightly underestimated the daily minimum 2-m temperature while CCLM-REF overestimated it. Consequently, the daily temperature range was overestimated by ICLM-REF and underestimated by CCLM-REF. However, the representation of the daily temperature range is, in general, in ICLM-REF closer to the observed values than in CCLM-REF. The overestimation of the diurnal range of temperature near the surface in ICLM-REF was probably caused by a positive radiation bias, which is known in ICON. This positive bias is especially large when the

radiation scheme RRTM (Rapid Radiative Transfer Model) (Mlawer et al., 1997; Barker et al., 2003) is used, which was the case in ICLM-REF (see Table 2). Recently, another radiation scheme, named ecRad (Rieger et al., 2019), is added into ICON. The positive bias for radiation is strongly reduced with the use of this radiation scheme.

ICLM-REF had also smaller RMSE than CCLM-REF for 2-m temperature over most sub-regions with one exception in sub-region Iberian Peninsula (Table 4). RMSEs of ICLM-REF did not exceed 0.87 K, whereas CCLM-REF had RMSE up to

1.03 K. Looking at the STDEV ratio, ICLM-REF had stronger spatial variation than E-OBS in three out of eight sub-regions and weaker variation in four out of eight. In some sub-regions, the STDEV ratio was equal or very close to one (France, Iberian Peninsula, Mediterranean). Compared with CCLM-REF, ICLM-REF showed in six sub-regions better STDEV ratios.

Similar results were found for minimum 2-m temperature, with RMSE from ICLM-REF between 0.36 and 0.85 K. All ICLM-REF's RMSEs were smaller compared to those from CCLM-REF, especially for sub-region Mediterranean and British

Isles ICLM-REF errors were less than half (Table 5). ICLM-REF's STDEV ratio was closer to 1 for all sub-regions in com-

parison with CCLM-REF. Both models did not simulate well the minimum 2-m temperature spatial variation in mountainous area (Scandinavia and Alps) as for flatter area, but did not show a tendency for specific type of orography.

Regarding RMSEs of maximum 2-m temperature (Table 6), ICLM-REF showed larger biases compared to min/mean 2-m temperature with a maximum bias of 1.54 K in the Mediterranean sub-region. In four out of eight sub-regions, ICLM-REF got lower errors and five sub-regions showed better spatial variation. Overall, ICLM-REF simulated average and daily max/min values of 2m air temperature better than CCLM-REF.

The improved representation of daily max/min 2-m temperature in ICLM-REF resulted in a reduced bias for climate indices which are determined by air temperature. Among those indices, CCLM-REF overestimated the annual numbers of ice days and tropical nights averaged over the evaluation period (1981-2000) as can be seen in Figure 11. Largest tropical night overestimation was found for the sub-region Eastern Europe with 5.2 nights per year by CCLM-REF, 6.5 times more than that of E-OBS (0.8 nights per year), while the ICLM-REF result was much closer to the observations with only 1.15 nights per year. Beside that, ICLM-REF tropical nights indices were very close to the observed numbers for sub-regions France, Alps, Mediterranean, while the numbers in CCLM-REF clearly stand out against the observations. The results for the annual ice days index were similar; CCLM-REF overestimated the number of ice days for all sub-regions. ICLM-REF, on the other hand, slightly underestimated the annual numbers of ice days but was in all sub-regions much closer to the number derived from E-OBS than CCLM-REF. Generally, ICLM-REF showed an underestimation of the annual frost days compared to observations over Europe, except for the Scandinavia sub-region. Compared to CCLM-REF, however, the underestimation was reduced in ICLM-REF. The biggest improvement was simulated in the sub-region Alps.

The number of annual summer days was overestimated by ICLM-REF for six from the eight sub-regions, while CCLM-REF mostly underestimated the amount of summer days. The strongest overestimation of the summer day index compared to CCLM-REF and E-OBS was seen in the Mediterranean sub-region. For three out of eight sub-regions and on average over the whole Europe, ICLM-REF simulated summer day indices more in agreement with observations compared to CCLM-REF. On average, for whole Europe, ICLM-REF resulted in 53.25 summer days per year; the numbers from E-OBS and CCLM-REF are 56.4 and 43.7, respectively.

## 5.2.2 Precipitation

The mean annual precipitation bias ranged mostly from -50 mm/month to 50 mm/month in both models (Figure 4). Overall, both models simulated more precipitation than E-OBS data. However, one should keep in mind that E-OBS precipitation data tend to suffer from gauge undercatch and evaporation leading to too low values (Gampe and Ludwig, 2017; Hofstra et al., 2009). That might be the reason why both ICLM-REF and CCLM-REF overestimated precipitation in large part of the domain. CCLM-REF tended to produce too little precipitation than E-OBS along the Atlantic coast while ICLM-REF agreed better with observation in this area. In the western part of Germany, for example, ICLM-REF had only a slight bias of 5 mm/month compared with the reference data. CCLM-REF, on the other hand, produced negative biases up to -20 mm/month in this area. However, over the eastern part of Germany, ICLM-REF had larger biases up to more than 10 mm/month. In all other regions, the annual spatial distribution of precipitation biases was quite similar in both models.

Looking at the spatial variability of the seasonal biases among the sub-regions in Figure 8, we see that although for some sub-regions in certain seasons, the bias medians were close to zero, the ranges of biases were large. This is expected because precipitation is a highly inhomogeneous variable. Summer and autumn tended to have small median bias in ICLM-REF and CCLM-REF. Among the four seasons, summer had the smallest variation probably due to the low precipitation amount. For winter, summer and autumn, it is not clear which model performed better. For spring, the median and the range of the bias were in better agreement with observations in CCLM-REF compared to ICLM-REF for most of the sub-regions.

RMSE for precipitation from ICLM-REF ranged from 10.48 to 30.52 mm (Table 7). The largest error appeared over the Alps sub-region, probably due to the complicated terrain and the dependency of precipitation on orography. RMSEs of CCLM-REF were smaller than those of ICLM-REF for most of the sub-regions, except for sub-region Iberian Peninsula and Scandinavia. ICLM-REF simulated larger variation of precipitation in space compared to E-OBS data, with most STDEV ratio larger than 1, except for British Isles with 0.8. The spatial variability of precipitation in CCLM-REF was closer to observations in six out of eight sub-regions indicated by an STDEV closer to one.

ICLM-REF tended to have more precipitation days than CCLM-REF, with the annual wet day index higher for most of the sub-regions and only one exception for sub-region Scandinavia (Figure 11). In six sub-regions, ICLM-REF was more in line with observation than CCLM-REF. ICLM-REF also produced more days with heavy and very heavy precipitation on yearly average than CCLM-REF. For most of the sub-regions, both models overestimated heavy and very heavy precipitation indices, but CCLM-REF was often closer to E-OBS. From our results, CCLM-REF performed better for precipitation and precipitation related climate indices. ICLM-REF had in part improvement in certain area of the model domain as well as for certain season and climate index.

### 5.2.3 Mean sea level pressure

The bias of MSLP of the two models showed opposite signs. ICLM-REF had positive biases while the biases in CCLM-REF were mostly negative as revealed in Figure 4. ICLM-REF had biases up to 6 hPa, while CCLM-REF had up to - 4 hPa. The performance of CCLM-REF was better, especially over Scandinavia and Iberian Peninsula. The opposite signs of the bias in the two model experiments can also be seen very clearly in Figure 9. Spring and autumn showed less spatial variability of MSLP values in both models. The representation of MSLP was in winter and autumn better in CCLM-REF than in ICLM-REF with smaller median bias and smaller bias ranges.

The RMSEs and STDEV ratios for MSLP are shown in Table 8. ICLM-REF had better RMSE than CCLM-REF in half of the sub-regions. However, ICLM-REF had some large errors over sub-regions Iberian Peninsula and France with RMSEs of 1.90 hPa and 1.71 hPa, respectively. Regarding spatial variation, ICLM-REF gave better STDEV ratio than CCLM-REF in most of the sub-regions. Both model resulted in largest difference in STDEV with respect to E-OBS data over sub-region Alps with the ratio of 1.76 for ICLM-REF and 2.09 for CCLM-REF.

Additionally, we evaluated the representation of the large-scale MSLP in ICON-CLM by looking at its spatial MSLP pattern. The driving data ERA-Interim was used for reference and ICLM-REF result was also compared to CCLM-REF. The mean spatial fields of MSLP from the three data sets were calculated for the evaluation period 1981-2000. All data were regridded

to ERA-Interim grid to ensure a fair comparison and are shown in Figure 12, the illustration areas are different among the data sets due to the different model domains. As can be seen, ICLM-REF could be able to reproduce the large-scale pattern of MSLP from the forcing data. The pressure structure looks quite similar between ICLM-REF and ERA-Interim, with the low and high pressure systems located in the right areas. Because of the small domain of ICLM-REF compared with ERA-Interim,

it is hard to view the whole high pressure system located west of Portugal. But from what we can see, the highest pressure was a bit closer to the coast line in ICLM-REF than in ERA-Interim. This unfortunately was out of CCLM-REF domain. ICLM-REF underestimated the MSLP near Iceland by several hectopascal. CCLM-REF reproduced the MSLP better than ICLM-REF in this area with the magnitude and pattern very similar to those from ERA-Interim.

### 5.2.4   Cloud cover

The representation of annual mean cloud cover in ICLM-REF looks much better than in CCLM-REF in Figure 4. CCLM-REF produced too much cloud over Scandinavia and over the eastern part of the domain (up to 0.2, or in percentage 20%, more cloud cover than CRU TS data). ICLM-REF, on the other hand, had biases of only up to +/- 10%. For most regions, the bias was in the range of +/- 5%. The overestimation of cloud cover in the sub-regions Scandinavia and Eastern Europe might be the reason for the cold bias of CCLM-REF in these regions (see Figure 4).

On seasonal mean as shown in Figure 10, however, CCLM-REF showed a smaller negative bias during winter in all sub-regions (except Scandinavia), and its performance in autumn was comparable to ICLM-REF. ICON-CLM simulated cloud cover noticeably better than COSMO-CLM over the sub-region Scandinavia.

The RMSEs of cloud cover did not show much difference between the two simulations as presented in Table 9. Same conclusion can be drawn for STDEV ratios, numbers were somewhat similar from both ICLM-REF and CCLM-REF.

### 5.3   Large-scale information reproduced by ICON-CLM

As the limited area models are only forced by the global data at their boundaries, inside the limited domain, the regional climate models can more or less freely develop their own circulation. One of the major concerns in regional climate modelling is to what extend the regional climate models modify the large-scale information from the global data (Sanchez-Gomez et al., 2009).

We want to test if ICON-CLM can be able to reproduce the large-scale atmospheric condition from the forcing data ERA-Interim. The geopotential at 500 hPa data from ICLM-REF and ERA-Interim were averaged at different temporal scales, 6 hourly, daily, monthly, seasonal and yearly. Due to the different grids and ERA-Interim lower resolution, ICLM-REF data were remapped to ERA-Interim grid. The spatial correlation coefficient was calculated for each time slot within the evaluation period 1981-2000.

Time series of the correlation coefficient for the different temporal scales are shown in Figure 13. For all of the time scales we considered, results showed higher correlation for longer time scale, the correlation is lowest for 6 hourly data and strongest for yearly data. For the 6 hourly and daily means, the mean correlation coefficient over the whole time period 1981-2000 was high, 0.925 and 0.928 respectively. But the correlation for some time slots dropped below 0.8, causing the minimum values

of correlation coefficient only over 0.5. ICON-CLM showed correlation coefficients above 0.85 for longer time scales from monthly to yearly. With this result, we can conclude that ICON-CLM reproduced well the large scale of the driving data for time scales from monthly to yearly, and partially for time scales from 6 hourly to daily.

## 6 Conclusions

The new regional climate model ICON-CLM has been derived from the weather forecast model ICON-LAM along with the necessary technical infrastructure and evaluation tools allowing users to carry out and evaluate long-term regional climate simulations. An evaluation run from the very first version of ICON-CLM showed very promising results. The ICON-CLM results were proven to be independent of the domain decomposition, and restarting with the same configuration gave binary identical results. ICON-CLM did not show any systematic dependency of the results on integration time steps which was found in COSMO-CLM. All tested time steps showed similar results with no bias outliers for any of the chosen values. These time step tests were done with model horizontal resolution R2B8 (about 10 km) over the EURO-CORDEX domain. When choosing another model grid spacing or simulating another domain, tests might be required to affirm these results.

The vertical nudging of the global forcing data at the model upper boundary did not show any notable impact on the climatology of surface variables. This is probably due to the fact that in our settings of ICON-CLM, the top of the atmosphere was at 30 km height. When choosing a lower model top, one might see larger effects of the vertical nudging. Furthermore, in this study, we focused on the near-surface climate and therefore did not look at the upper air layers where larger differences between nudging and no nudging can be found.

Results from the evaluation run showed that ICON-CLM was able to reproduce the large scale atmospheric circulation from the driving model and the most important climate variables. In comparison with reference data, ICLM-REF biases are of similar magnitude, for certain areas or certain variables, even slightly smaller than CCLM-REF biases. Improvements were visible in ICLM-REF for air temperature, and hence temperature-related climate indices. The reason might be that ICLM-REF showed a lower overestimation of cloud coverage over the northern and eastern part of the domain compared to CCLM-REF. For precipitation and MSLP, the results from CCLM-REF were in better agreement with E-OBS data.

It should be taken into account that ICON-CLM is still in the early stage of development in climate mode while COSMO-CLM has been developed and applied for NWP and climate applications for more than 20 years and was well-tuned with a large number of tested namelist combinations in the COPAT project. Therefore, ICON-CLM has still great potential to improve its model setup and with growing experience we expect that ICON-CLM results will improve further in the upcoming years.

Comparison for air temperature between ICLM-REF and the EURO-CORDEX ensemble showed that ICLM-REF biases landed in the upper part of the ensemble bias spreads. A broader inter-comparison with EURO-CORDEX ensemble is recommended once an optimum set-up for ICON-CLM is established.

As written above, only short tests were done on a different computing system other than the Cray XC 40 at DWD. More technical tests, therefore, should be done on the Atos/Bull at DKRZ. Also a comparison of the results from different machines will show how ICON-CLM is dependent on the computer systems.

The next step in the ICON-CLM preparation will be a thorough model tuning by testing the sensitivity of the model to a variety of namelist parameters and different combinations of namelist settings in order to find an optimal configuration. Climate simulations on different domains, e.g. CORDEX Africa, will be done to evaluate the ability of ICON-CLM to simulate different climates. So far only re-analysis driven simulations have been performed with ICON-CLM, but historical simulations driven by the results of global climate simulations will also be done to test the model performance for this experiment type. Based on a well-evaluated model configuration, regional climate projections will be carried out in order to address the impact of climate change at regional scale. Thus, climate projections, e.g. in the framework of CORDEX, will also be provided with ICON-CLM in the future. We plan to further develop ICON with the aim of unifying the different physics packages currently existing for the numerical weather prediction and the climate mode in order to pursue a "Seamless prediction" system with one forecasting system which can produce forecasts and projections for all time-scales from weather prediction to seasonal, decadal predictions and climate projections.

*Code availability.* To institutions, the ICON model is distributed under an institutional license issued by DWD. In case of the institutional license, two copies of the institutional license need to be signed and returned to the DWD. The ICON Release Version 2.6.1 can be then downloaded at https://data.dwd.de.

To individuals, the ICON Model is distributed under a personal non-commercial research license distributed by MPI-M. Every person receiving a copy of the ICON framework code accepts the ICON personal-non-commercial research license by doing so. Or, as the license states: Any use of the ICON-Software is conditional upon and therefore leads to an implied acceptance of the terms of the Software License Agreement. To receive an individually licensed copy, please follow the instructions provided at https://code.mpimet.mpg.de/projects/iconpublic/wiki/Instructions_to_obtain_the_ICON_model_code_with_a_personal_non-commercial_research_license. The ICON Release Version 2.6.1 model code with a personal non-commercial research license can be obtained at https://code.mpimet.mpg.de/projects/icon-downloads/files.

The starter package ICON-CLM_SP_beta1 can be downloaded from http://doi.org/10.5281/zenodo.3896136. The starter package will be shipped with a recommended configuration for EURO-CORDEX domain at the resolution R2B8 and will be available for all CLM-Community members.

The forcing data were used for our simulations are the ERA-Interim data (Dee et al., 2011). Model evaluation was done againt E-OBS version 10.0 and 12.0 (Haylock et al., 2008; Van den Besselaar et al., 2011) und CRU TS version 3.22 datasets (Harris et al., 2014).

**Appendix A**

**A1**

*Author contributions.* Trang Van Pham implemented the adaptations in model code and ICON-CLM Starter Package, carried out the experiments and prepared the manuscripts with contributions from all co-authors. Christian Steger contributed in the organization and supervision of the work. Ingo Kirchner and Mariano Mertens are the source code administrators of ICON-CLM. Burkhardt Rockel contributed to the

ICON-CLM Starter Package. Klaus Keuler, Burkhardt Rockel and Barbara Früh defined the research goals, aims and methodology. Daniel Rieger and Günther Zängl contributed in the development of ICON-NWP, the foundation for ICON-CLM.

*Competing interests.* The authors declare that they have no conflict of interest.

*Disclaimer.* TEXT

5 *Acknowledgements.* The authors thank to all ICON development teams at DWD and MPI-M for providing us the ICON model code which is a great basis for our development. Thanks to the colleagues who provided the COSMO-CLM evaluation from COPAT project. Trang Van Pham acknowledged the financial support of DWD via project ProWaS. We acknowledged the E-OBS dataset from the EU-FP6 project ENSEMBLES (http://ensembles-eu.metoffice.com) and the data providers in the ECA&D project (http://www.ecad.eu)"

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

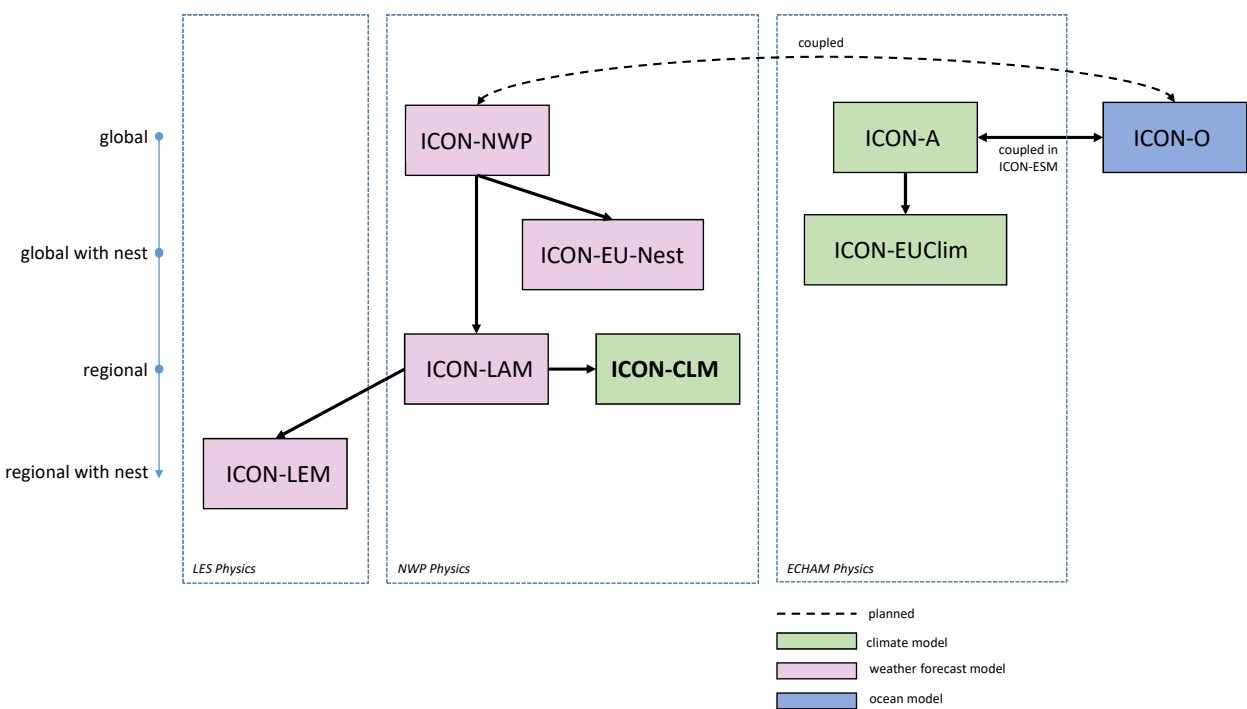

**Figure 1.** Overview ICON modelling framework.

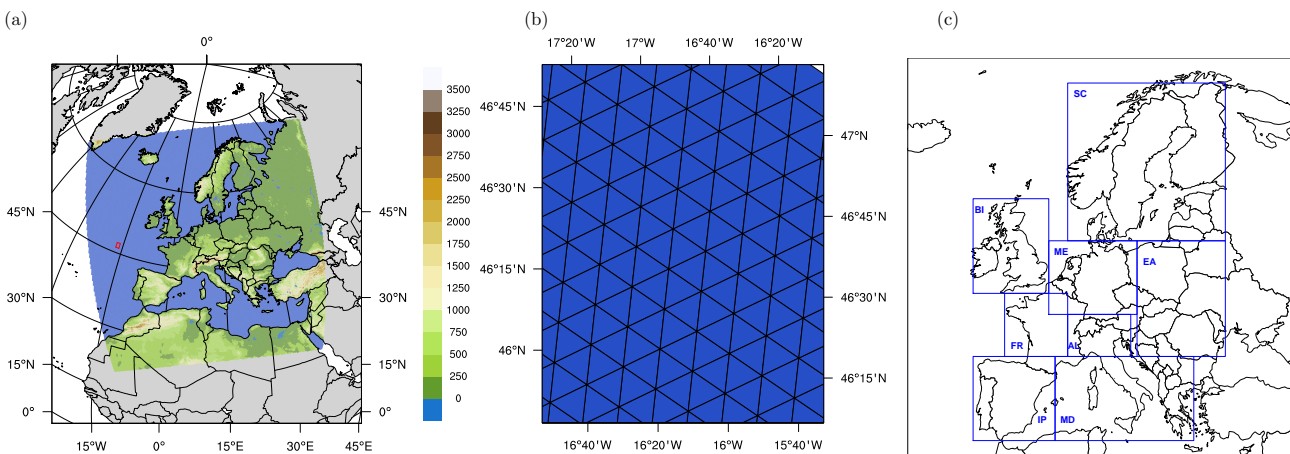

**Figure 2.** (a) Simulation domain EURO-CORDEX and model orography [m] of ICLM-REF on the R2B8 grid. (b) Illustration of the icosahedral grid of ICON-CLM at the resolution R2B8 from a closer look at the red marked region from sub-figure (a). (c) Evaluation was done for the eight PRUDENCE sub-regions (BI: British Isles, IP: Iberian Peninsula, FR: France, ME: Mid-Europe, SC: Scandinavia, AL: Alps, MD: Mediterranean and EA: Eastern Europe).

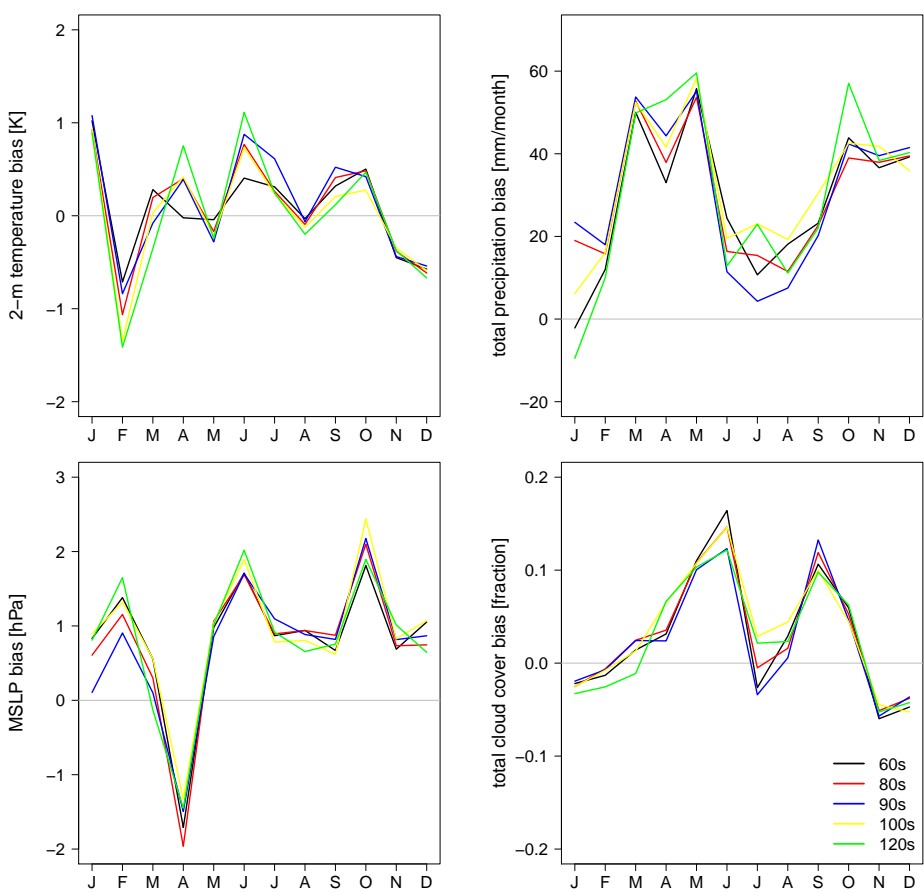

**Figure 3.** Monthly averaged biases from ICON-CLM simulations with different time steps for 2-m temperature, total precipitation, MSLP compared to E-OBS data and for total cloud cover compared with CRU TS data. Data were averaged for the Alps (AL) sub-region and for year 1979.

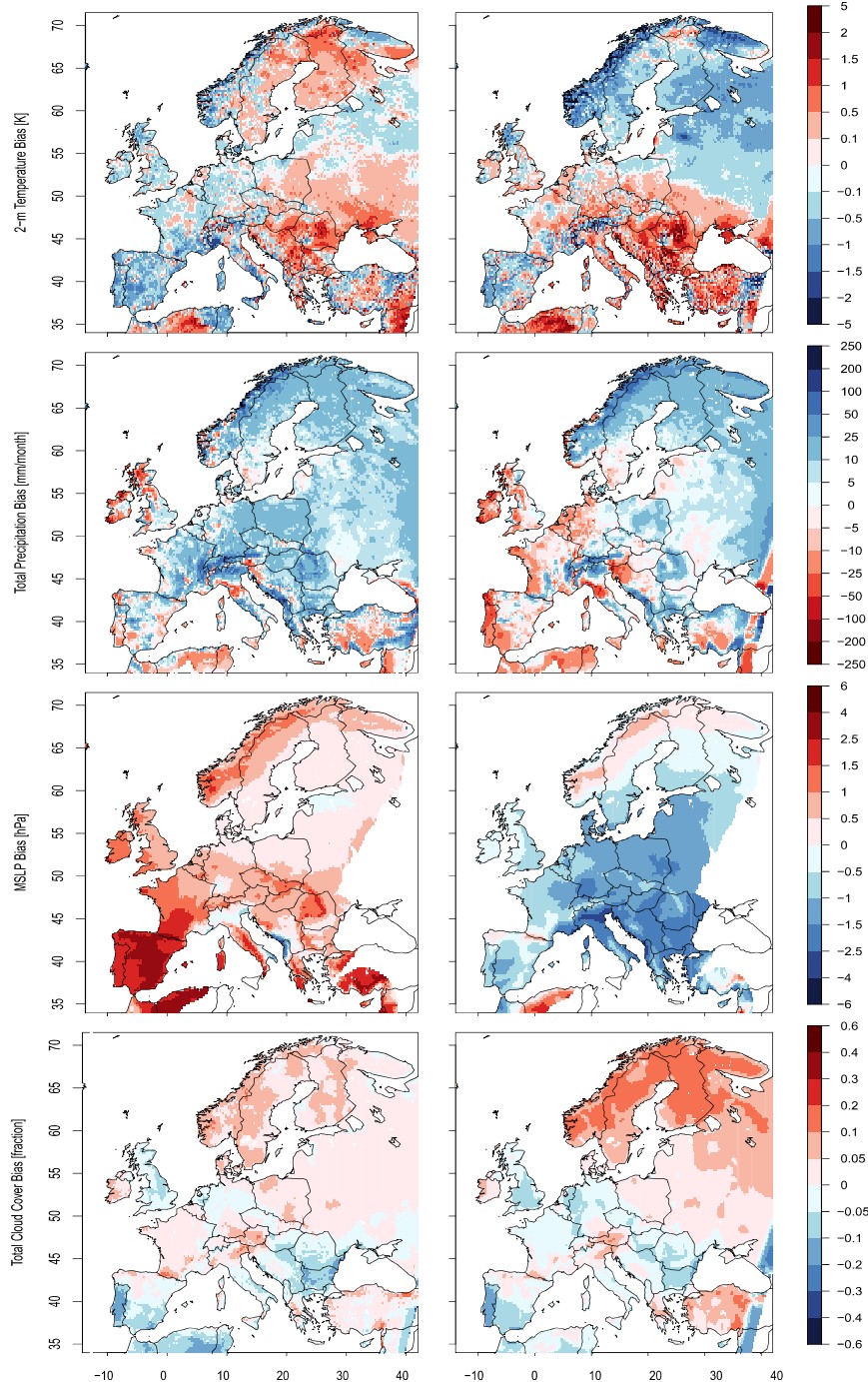

**Figure 4.** Multi-year averaged biases over the period 1981-2000 against E-OBS data for 2-m temperature, total precipitation, MSLP and against CRU TS data for total cloud cover (from top to bottom respectively). Data from ICLM-REF and CCLM-REF evaluation runs (left and right respectively).

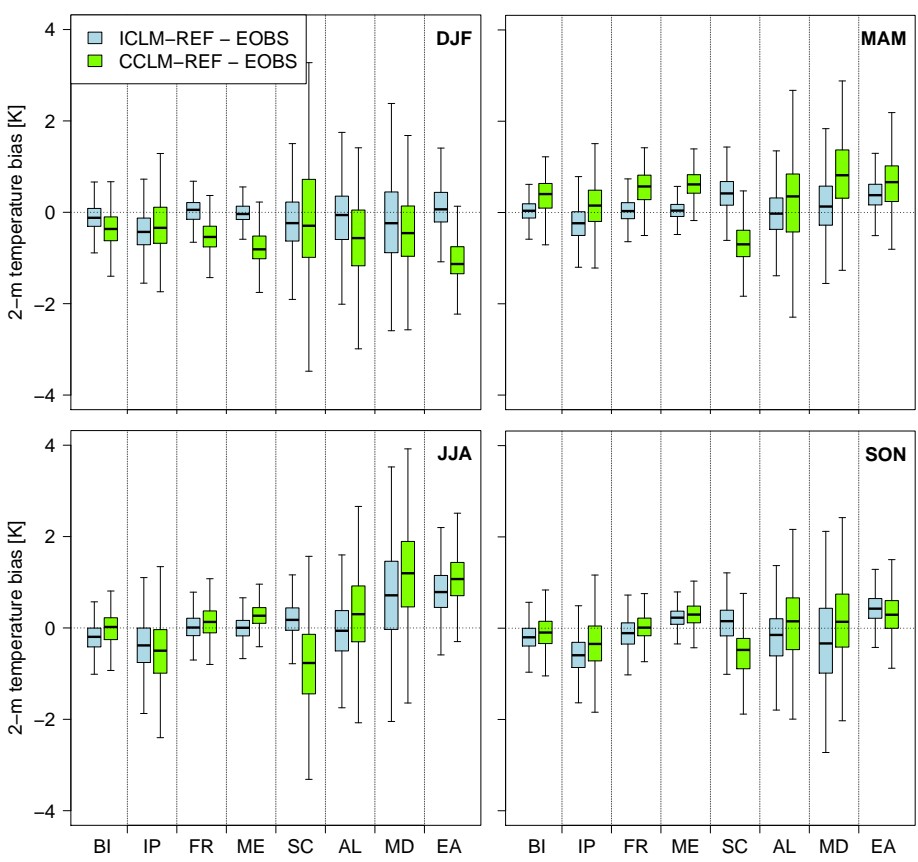

**Figure 5.** Seasonal mean 2-m temperature biases of ICLM-REF and CCLM-REF data against E-OBS data for all PRUDENCE sub-regions. Values averaged for the period 1981 to 2000. Spatial variability within a sub-region is indicated by the lower bar (5th), upper bar (95th), lower edge of the box (25th), middle of the box (50th) and upper edge of the box (75th percentile of the distribution of all grid points within a sub-region).

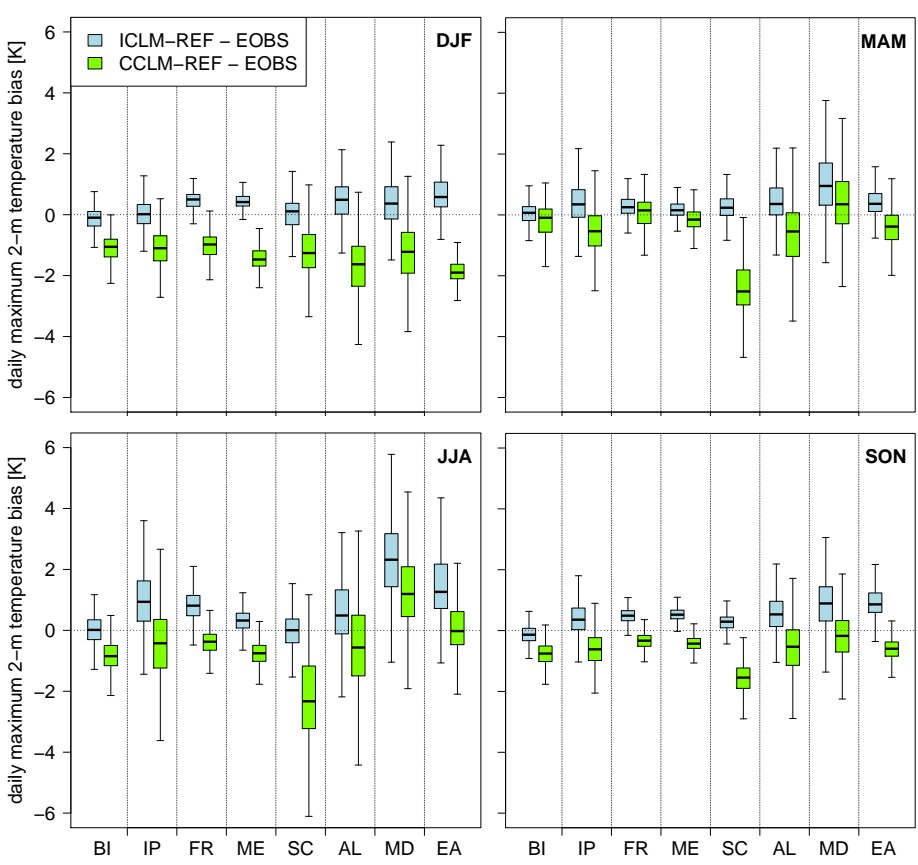

**Figure 6.** Same as Figure 5 but for daily maximum 2-m temperature.

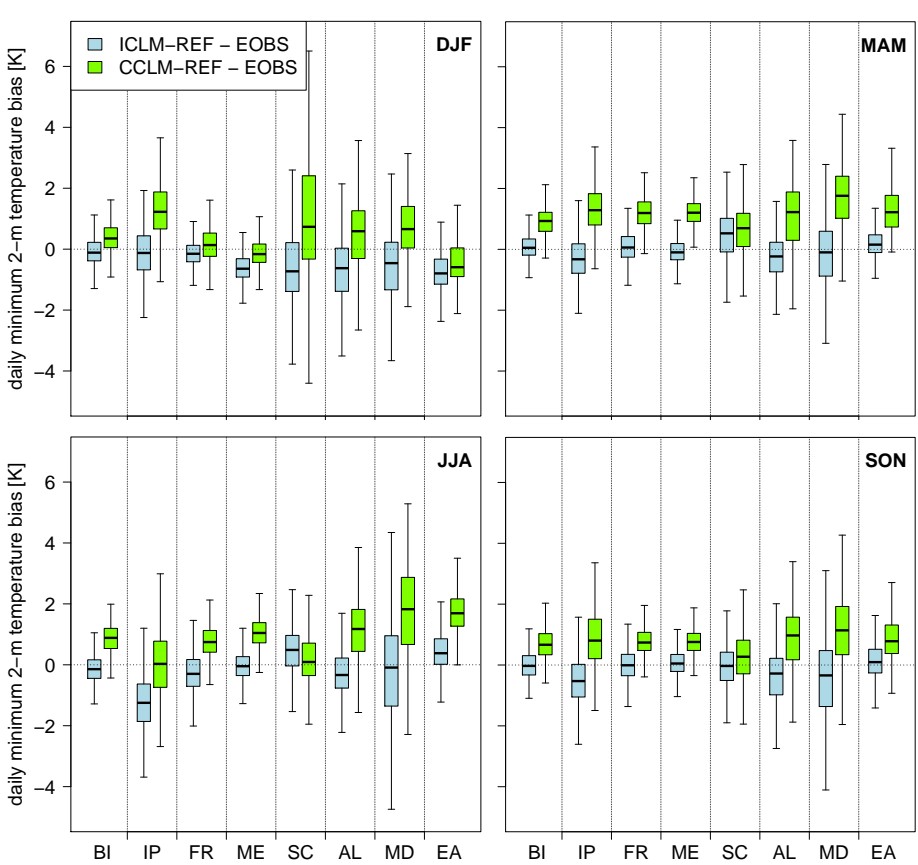

**Figure 7.** Same as Figure 5 but for daily minimum 2-m temperature.

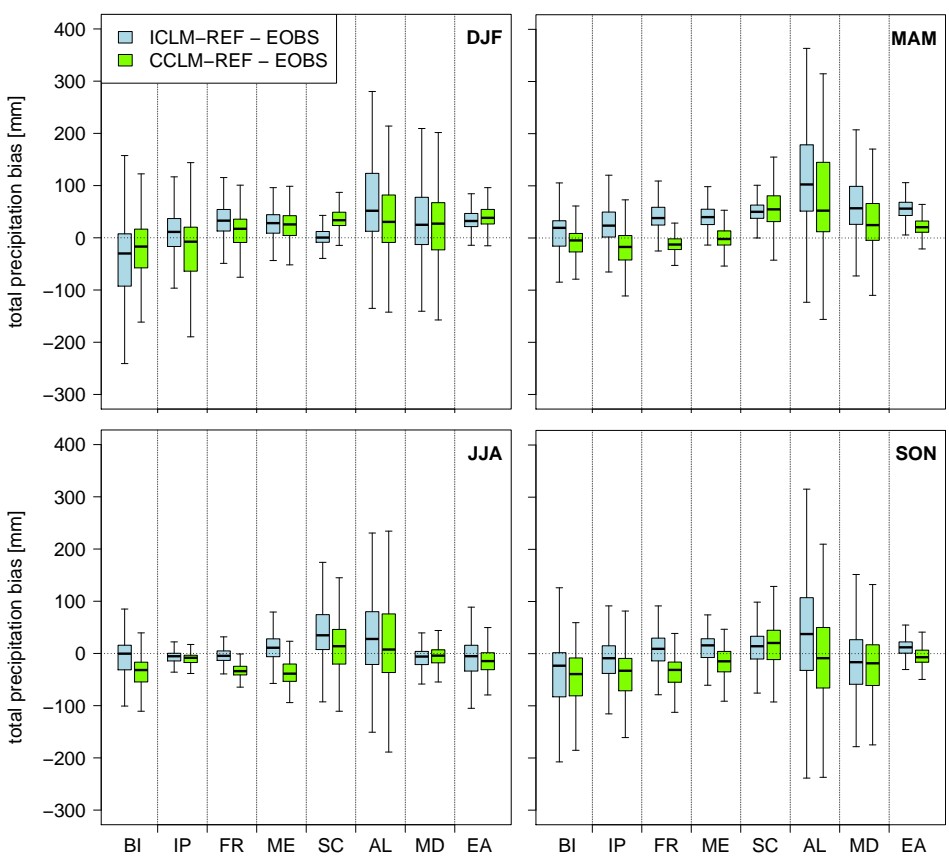

**Figure 8.** Same as Figure 5 but for total precipitation.

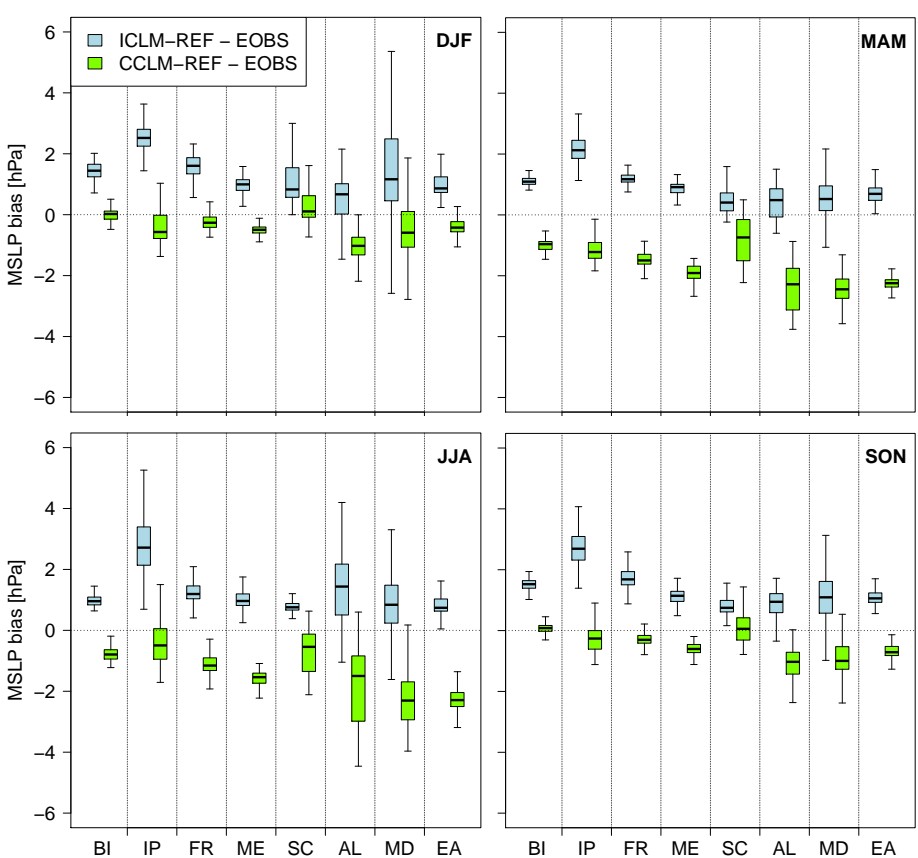

**Figure 9.** Same as Figure 5 but for MSLP.

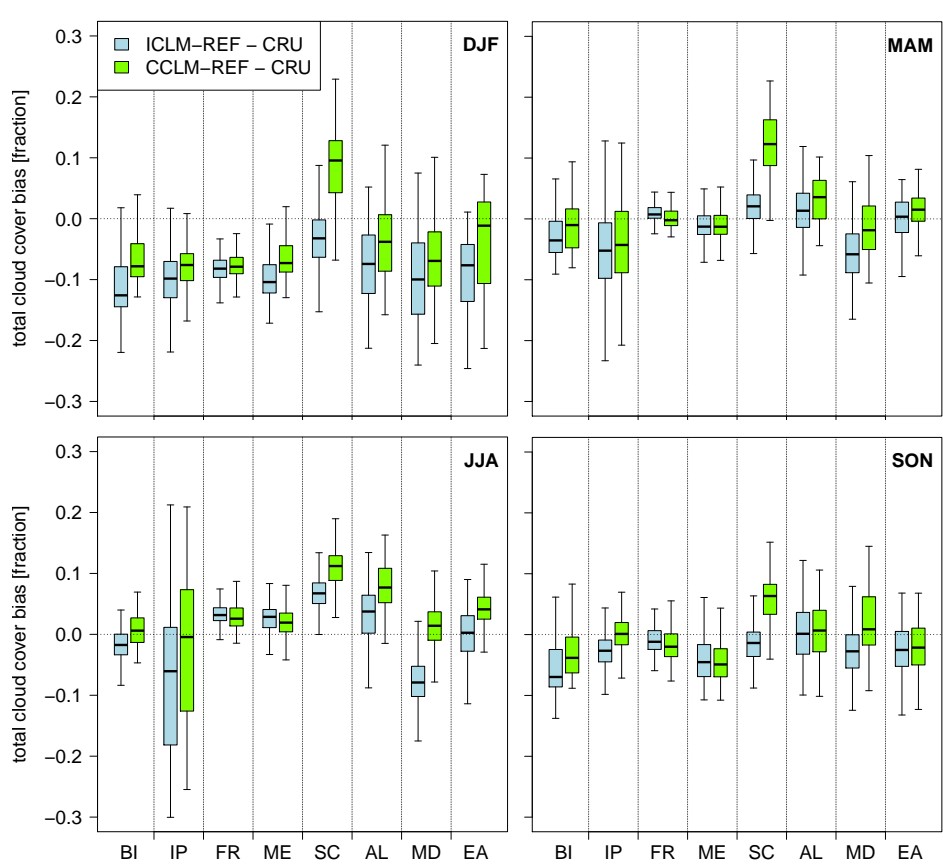

**Figure 10.** Same as Figure 5 but for total cloud cover.

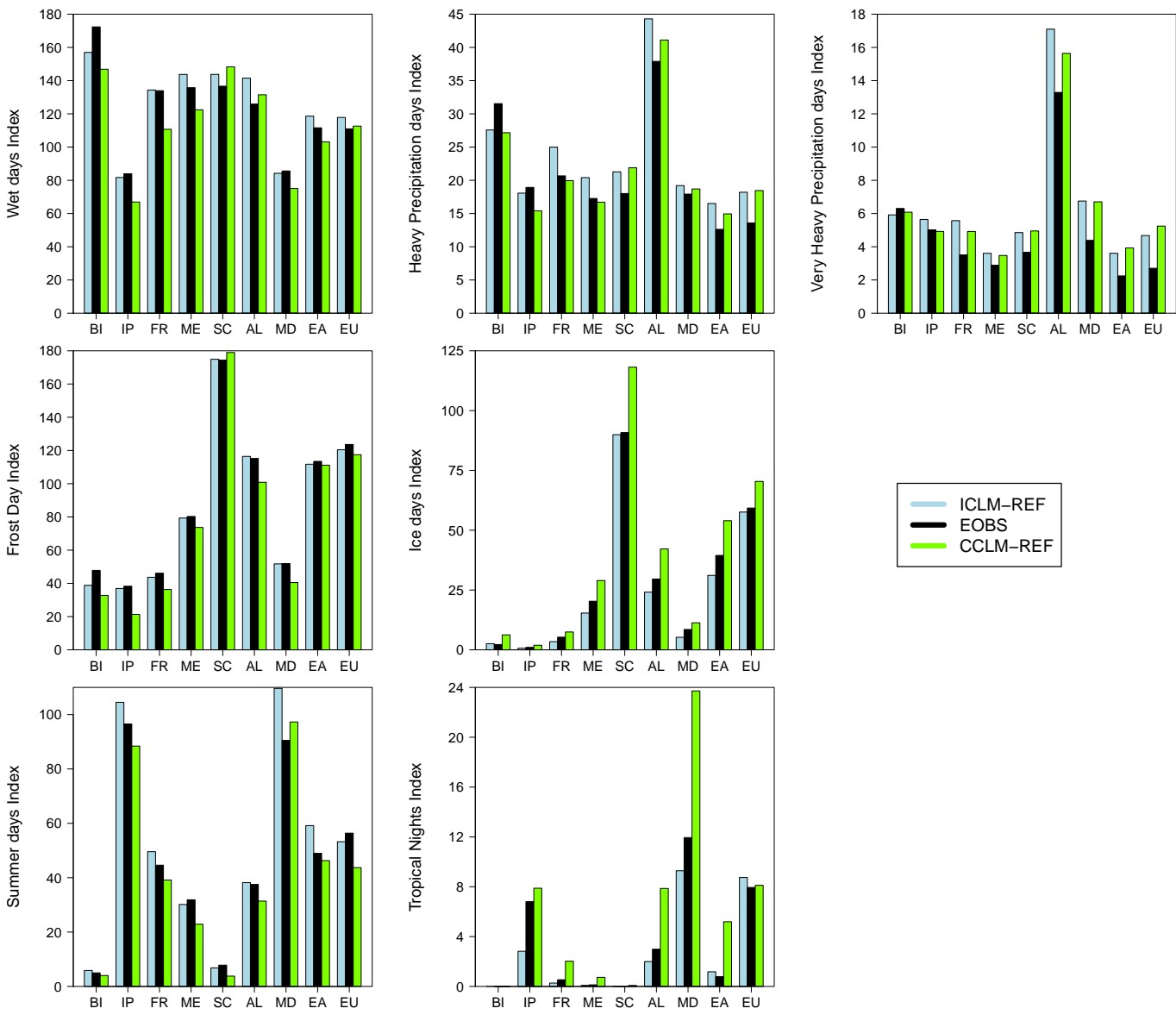

**Figure 11.** Area mean of climate indices [days per year] calculated from daily maximum and minimum air temperature and daily precipitation data of ICLM-REF, CCLM-REF and E-OBS for the PRUDENCE sub-regions and for the whole Europe (denoted EU) averaged over the period 1981-2000.

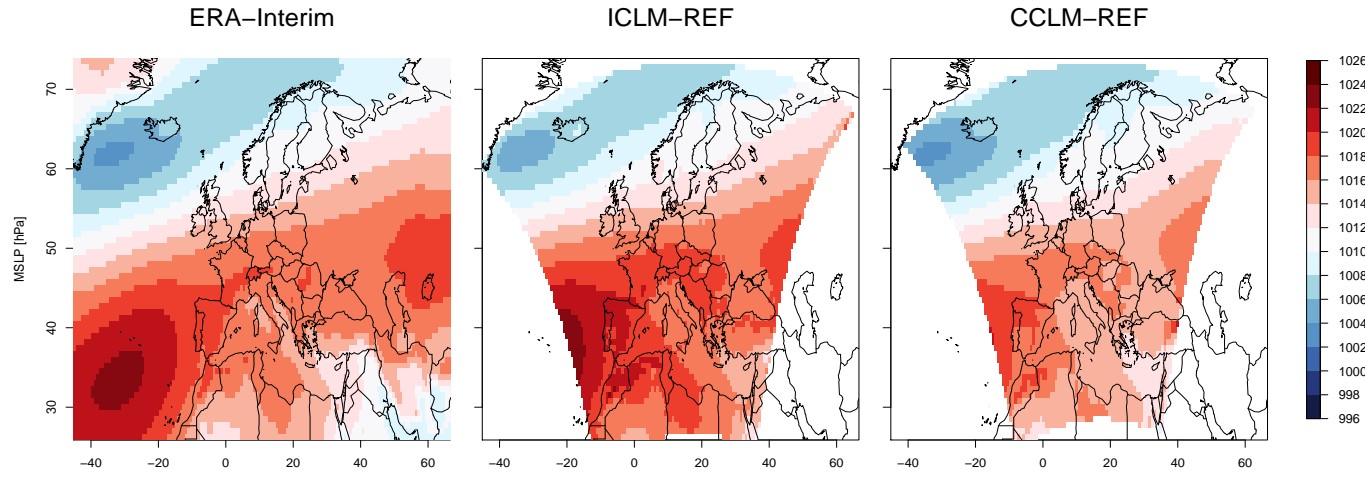

**Figure 12.** Annual mean sea level pressure (MSLP, in hPa) from ERA-Interim (left), ICLM-REF (middle) and CCLM-REF data (right). Data were averaged over the period 1981-2000.

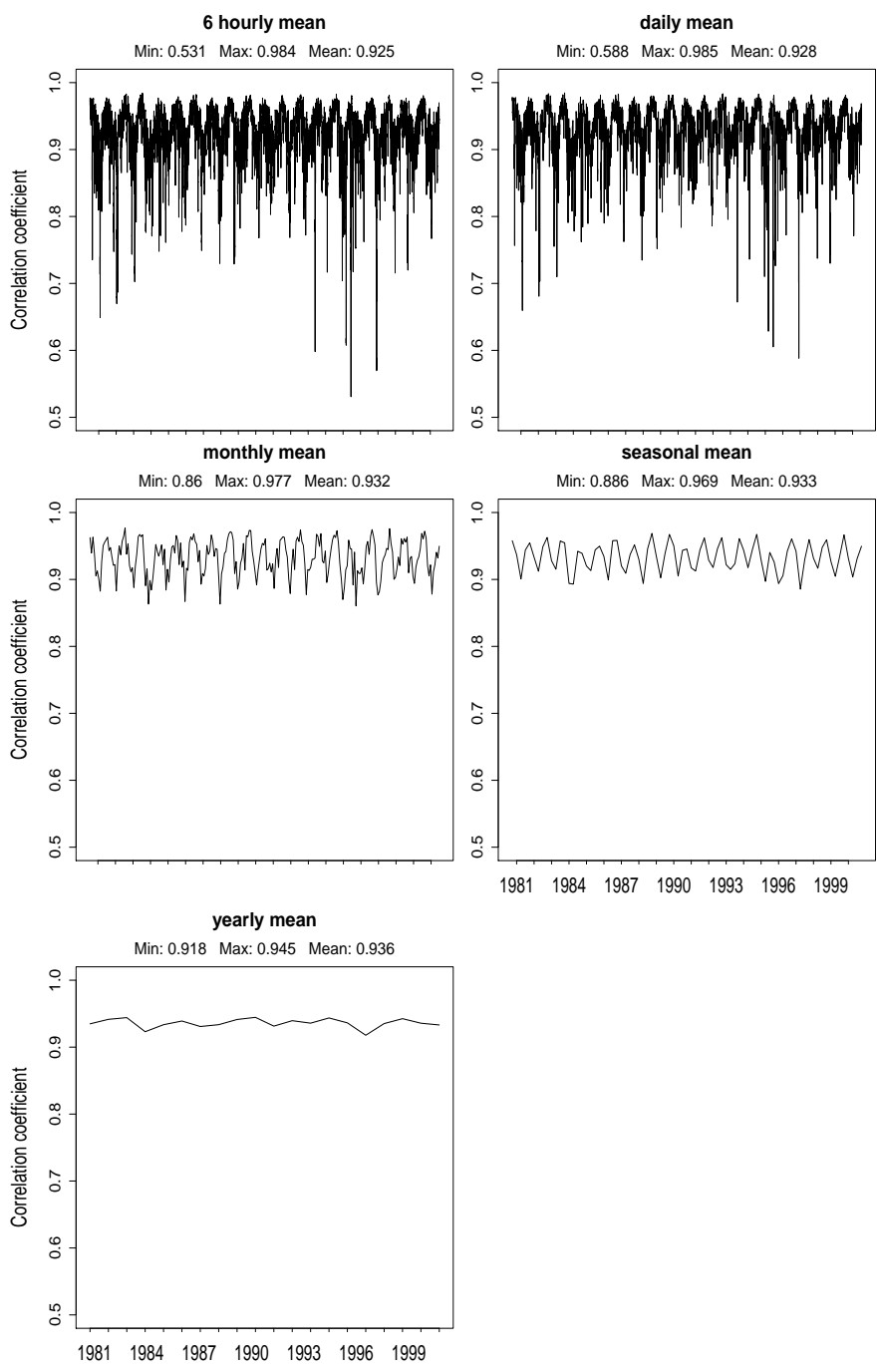

**Figure 13.** Spatial correlation coefficient of geopotential at 500 hPa between ICLM-REF and ERA-Interim data for various temporal scales, 6 hourly, daily, monthly, seasonal and yearly. Data were taken in the period 1981-2000.

**Table 1.** Total numbers of triangle cells, edges and effective grid resolution of the grids mentioned in this paper.

| Grid | Domain | Number of triangle cells | Number of triangle edges | Effective grid resolution [km] |
|------|--------|--------------------------|--------------------------|--------------------------------|
| R2B8 | EURO-CORDEX | 286 824 | 430 988 | 9.9 |
| R3B7 | global | 2 949 120 | 4 423 680 | 13.2 |
| R3B8 | Europe nested in global R3B7 | 659 156 | 989 911 | 6.6 |

**Table 2.** Comparison between COSMO-CLM and ICON-CLM models.

| | COSMO-CLM | ICON-CLM |
|---|---|---|
| Planetary boundary layer scheme | Prognostic turbulent kinetic energy closure<br>Doms et al. (2011) | Wave dissipation at critical level<br>Orr et al. (2010) |
| Cumulus convection scheme | Shallow convection: Reduced Tiedtke scheme for shallow convection only (Tiedtke, 1989)<br>Deep convection: Tiedtke (1989) oder Tiedtke-Bechtold scheme from IFS Model | Mass flux scheme with shallow, deep or mid-level convection.<br>CAPE (convective available potential energy) based closure for deep convection.<br>Boundary layer equilibrium closure for shallow convection.<br>Large-scale omega (vertical velocity in pressure coordinates) based closure for mid-level convection.<br>Tiedtke (1989); Bechtold et al. (2008) |
| Cloud microphysic scheme | Seifert and Beheng (2001), reduced to one-moment scheme | Single-moment scheme<br>Doms et al. (2011); Seifert and Beheng (2001) |
| Radiation short and long wave | $\delta$ two-stream radiation scheme after Ritter and Geleyn (1992) | RRTM (Rapid Radiative Transfer Model) (Mlawer et al., 1997; Barker et al., 2003) |
| Land surface and soil scheme | TERRA-ML<br>Doms et al. (2011) | Tiled TERRA<br>Schrodin and Heise (2001); Schulz et al. (2016) |
| Coordinate system | – horizontal: rotated geographical (lat/lon)<br>– vertical: terrain following Gal-Chen height coordinate (Gal-Chen and Somerville, 1975) and exponential height coordinate (SLEVE) according to Schär et al. (2002) | – horizontal: icosahedral grids<br>– vertical: terrain following Gal-Chen height coordinate (Simmons and Burridge, 1981) and exponential height coordinate (SLEVE) according to Schär et al. (2002) and Leuenberger et al. (2010) |

**Table 3.** Description of climate indices.

| Index | Description | Unit |
|---|---|---|
| Frost days index | Number of days with minimum 2-m temperature $< 0^\circ$C | Days |
| Ice days index | Number of days with maximum 2-m temperature $< 0^\circ$C | Days |
| Summer days index | Number of days with maximum 2-m temperature $> 25^\circ$C | Days |
| Tropical nights index | Number of days with minimum 2-m temperature $> 20^\circ$C | Days |
| Wet days index | Number of days with total precipitation $\geq 1$ mm | Days |
| Heavy precipitation days index | Number of days with total precipitation $> 10$ mm | Days |
| Very heavy precipitation days index | Number of days with total precipitation $> 20$ mm | Days |

**Table 4.** RMSE and spatial STDEV ratio of averaged monthly 2-m temperature for the PRUDENCE sub-regions. Data from ICLM-REF, CCLM-REF and E-OBS from 1981 to 2000.

| | RMSE [K] | | Standard deviation ratio | |
|---|---|---|---|---|
| | ICLM-REF | CCLM-REF | ICLM-REF/EOBS | CCLM-REF/EOBS |
| BI | 0.34 | 0.39 | 1.08 | 1.16 |
| IP | 0.54 | 0.44 | 0.99 | 0.97 |
| FR | 0.53 | 0.56 | 1.00 | 0.98 |
| ME | 0.56 | 0.73 | 0.99 | 0.98 |
| SC | 0.60 | 0.76 | 0.97 | 1.02 |
| AL | 0.53 | 0.59 | 1.03 | 1.08 |
| MD | 0.65 | 0.87 | 0.92 | 0.92 |
| EA | 0.87 | 1.03 | 1.10 | 1.12 |

**Table 5.** Same as Table 4 but for minimum 2-m temperature.

| | RMSE [K] | | Standard deviation ratio | |
|---|---|---|---|---|
| | ICLM-REF | CCLM-REF | ICLM-REF/EOBS | CCLM-REF/EOBS |
| BI | 0.36 | 0.87 | 1.03 | 1.08 |
| IP | 0.81 | 1.15 | 0.96 | 0.88 |
| FR | 0.69 | 0.95 | 0.96 | 0.89 |
| ME | 0.72 | 1.05 | 1.06 | 0.93 |
| SC | 0.81 | 0.84 | 0.90 | 0.89 |
| AL | 0.75 | 1.06 | 1.13 | 1.19 |
| MD | 0.60 | 1.53 | 0.94 | 0.90 |
| EA | 0.85 | 1.38 | 1.03 | 1.07 |

**Table 6.** Same as Table 4 but for maximum 2-m temperature.

| | RMSE [K] | | Standard deviation ratio | |
|---|---|---|---|---|
| | ICLM-REF | CCLM-REF | ICLM-REF/EOBS | CCLM-REF/EOBS |
| BI | 0.42 | 0.88 | 1.20 | 1.25 |
| IP | 0.81 | 0.81 | 1.02 | 1.08 |
| FR | 0.86 | 0.77 | 1.01 | 1.00 |
| ME | 0.76 | 1.01 | 1.06 | 1.09 |
| SC | 0.58 | 1.91 | 1.02 | 1.12 |
| AL | 0.80 | 1.11 | 0.99 | 1.03 |
| MD | 1.54 | 1.08 | 1.05 | 1.05 |
| EA | 1.29 | 1.19 | 1.18 | 1.12 |

**Table 7.** Same as Table 4 but for monthly accumulated total precipitation.

| | RMSE [mm] | | Standard deviation ratio | |
|---|---|---|---|---|
| | ICLM-REF | CCLM-REF | ICLM-REF/EOBS | CCLM-REF/EOBS |
| BI | 16.78 | 16.03 | 0.80 | 0.80 |
| IP | 10.48 | 11.85 | 1.11 | 0.94 |
| FR | 16.67 | 14.23 | 1.42 | 1.18 |
| ME | 13.72 | 12.43 | 1.09 | 0.87 |
| SC | 14.77 | 14.88 | 1.15 | 1.08 |
| AL | 30.52 | 22.53 | 1.38 | 1.27 |
| MD | 17.32 | 12.36 | 1.66 | 1.47 |
| EA | 15.91 | 12.19 | 1.30 | 1.06 |

**Table 8.** Same as Table 4 but for MSLP.

| | RMSE [hPa] | | Standard deviation ratio | |
|---|---|---|---|---|
| | ICLM-REF | CCLM-REF | ICLM-REF/EOBS | CCLM-REF/EOBS |
| BI | 1.60 | 0.91 | 1.02 | 0.94 |
| IP | 1.90 | 0.77 | 1.19 | 1.29 |
| FR | 1.71 | 1.12 | 1.38 | 1.11 |
| ME | 1.43 | 1.56 | 1.04 | 0.91 |
| SC | 1.21 | 0.91 | 0.93 | 0.74 |
| AL | 1.32 | 1.87 | 1.76 | 2.09 |
| MD | 1.68 | 1.79 | 0.74 | 0.73 |
| EA | 1.44 | 1.91 | 1.21 | 0.93 |

**Table 9.** Same as Table 4 but for total cloud cover.

| | RMSE [fraction] | | Standard deviation ratio | |
|---|---|---|---|---|
| | ICLM-REF | CCLM-REF | ICLM-REF/CRU | CCLM-REF/CRU |
| BI | 0.08 | 0.06 | 1.00 | 1.00 |
| IP | 0.10 | 0.09 | 1.33 | 1.33 |
| FR | 0.07 | 0.08 | 1.00 | 0.75 |
| ME | 0.07 | 0.07 | 0.33 | 0.67 |
| SC | 0.06 | 0.10 | 1.33 | 1.67 |
| AL | 0.07 | 0.08 | 1.00 | 1.00 |
| MD | 0.08 | 0.08 | 0.83 | 0.83 |
| EA | 0.07 | 0.06 | 1.67 | 1.67 |