# Peer review of "ICON in Climate Limited-area Mode (ICON Release Version 2.6.1): a new regional climate model"

_Geoscientific Model Development, 2020_

## Referee Comment (RC1) · Anonymous Referee #1 · 4 Aug 2020

The manuscript by Van Pham et al. presents a first reanalysis-driven evaluation run of the new ICON-CLM regional climate model which originates from the limited-area version of the global ICON model. The performance of the new modelling system is compared to the one of COSMO-CLM, the regional model previously employed by the German Weather Service and the CLM-Community. Although ICON-CLM is a comparatively new development and options for model calibration have probably not been fully exploited yet, its performance is comparable and partly even superior to the well-tested recommended COSMO-CLM configuration.

Overall, the work certainly fits into the journal's scope. One can expect ICON-CLM

to become one of the major RCMs employed in Europe, and the documentation and comparison of its performance is an important part of scientific progress. The paper is well written and well structured, data and methods are well explained. The quality of the figures is acceptable for most parts. The presentation of the results is rather descriptive, which is however acceptable for this kind of work. The conclusions are well based on the results obtained and the use of the English language is acceptable (though it could still be improved).

I can therefore recommend a publication of this work after a couple of minor issues have been fixed (see below). With kind regards.

MINOR ISSUES

- Model naming conventions and use of abbreviations. The reader of the paper easily gets confused with the names of the modelling systems (COSMO, COSMO-CLM, ICON-CLM, ICON-A, ICON-NWP, ICON-LAM, ICON-O, ICON-EUclim, ICON-EU-Nest, ICON-GCM etc). This is in party unavoidable as the names reflect the complex history of the ICON model. Figure 1 is certainly helpful in this respect, but as a reference I would appreciate an additional table that lists the individual configurations of ICON and their basic characteristics. Such a table could also serve as a reference for further publications. Furthermore, the paper also introduces the abbreviations ICLM-REF and CCLM-REF, which do not denote model systems but individual simulations but could well be confused with the model system names. I believe this could somehow be simplified, although – frankly - I currently have no idea how. One possibility might be to mention the simulation names in the suggested table as well. In any case, the reference to the simulations is not consistent throughout the paper, especially within the figures. Figure 3, for instance uses "ICLM-EOBS" in the headers, which should actually read "ICL-REF – EOBS" (same for CRU). Same for Figures 5 to 10. Figure 4 just uses "ICLM" and "CCLM" as headers (should be "ICLM-REF" and "CCLM-REF"). Also Tables 5 to 7 use "ICLM" and "CCLM" instead of "ICL-REF" and "CCLM-REF" in the headings.

- page 1 line 10: "with the setups similar

- page 2 lines 9-10: It is not really clear to me whether ICON-EU-Nest is a global model with regional refinement over Europe or a higher-resolution version traditionally nested into a global ICON model.

- page 2, line 10: "COSMO-EU" needs an explanation

- page 2 lines 13-14: As written, the unification was scheduled for end of 2019, which is already in the past. The sentence should hence be modified to "The implies that the last unification of COSMO and COSMO-CLM (COSMO 6), carried out at the end of 2019, was the last one."

- page 2 line 20: Wouldn't it be better to speak of "climate projection" here instead of "climate prediction" to highlight the longest time horizon for application of the model?

- page 2, line 23: Did ICON-NWP inherit any parts from COSMO (for instance, the physics package)?

- page 2 line 34: Why was CCLM 5.0 clm9 used for the comparison and not the latest (and final) unified version COSMO 6 (see above)?

- page 3 lines 13-14: Could you briefly explain why this is the case?

- page 3 lines 20-21: I suggest to replace this expression by "... time dependent GHGs as provided by corresponding GHG scenarios".

- page 5 line 13: It is not really clear which "necessary changes" are meant here.

- page 7 line 12: How were the data transformed/regridded?

- page 7 line 18: Would be better to speak of "ICLM-REF" and "CCLM-REF" here as the simulations themselves are meant.

- page 7 line 28: "a very good performance" -> this statement actually needs some quantification or should, alternatively, be reformulated

- page 7 lines 28-30: This paragraph actually summarizes the results described later on. It should not precede the presentation of the results I believe, but should rather be moved to the end of the results chapter or even to the conclusions chapter.

- page 8 line 17: "trends" is misleading here. I'd rather speak of "biases" or "results".

- page 8 lines 17-18: Any ideas WHY?

- page 9 line 5: "over the whole evaluation period" -> this expression is misleading as the figure shows no time series of the bias.

- page 9 lines 15-16: The numbers obviously refer to events per 20 years. Without providing the length of the period the numbers are, however, not interpretable. I'd suggest to use the unit [days per year] for these numbers and, actually, for the entire Figure 11. This is much easier to understand and to compare to other studies.

- page 9 line 20: "too low values" -> you're obviously referring to gauge undercatch and evaporative losses here, this should be mentioned (and supported by some reference).

- MSLP evaluation in Chapter 4.3: MSLP is evaluated in the same fashion as the other variables, but I see rather little value in this. What is most important here is the spatial MSLP pattern (in addition to the general magnitude), so the evaluation should consider the mean spatial field. The authors might think about replacing their MSLP evaluation by some more informative MSLP analysis.

- Figure 17: In addition to the naming of the simulations (see above) the variable names in the headers refer to the internal model names of the respective parameters. This is rather cryptic and could be replaced by the full names (2m temperature, cloud cover etc). Alternatively, the abbreviations should be mentioned in the caption.

- Figure 4: These maps obviously employ some uncommon projection and the European continent seems a little "distorted". Why don't you use the same projection as in Figure 2, for instance?

[Figure]

- Figures 8 and 9: The y-axis of these figures misses a label and the unit of the bias

- Figure 11: As explained above I'd suggest to use the unit [days per year] or [events per year] for these indicators (instead of [days per 20 years]). Furthermore, in to-pographically structured sub-domains such as AL or SC, the spatial averaging of the number of days defined by a temperature threshold makes little sense in my opinion as it completely neglects the large spatial variability. One way to improve on this might be to present the bars as medians with whiskers on top of it reflecting P5 and P95 of the spatial variability within a subdomain. Furthermore, I suggest to place the black EOBS bars to the left of the green simulation bars, not between them (also modify the legend in this case)

- Table 1 is hardly readable, the space between the table lines should be enlarged

SPELLING AND GRAMMAR

- page 1 line 7: "tests"

- page 2 line 1: "the CLM-Community developed"

- page 3 line 16: "with a user-defined"

- page 3 line 19: "the greenhouse gas"

- page 3 line 23: "retrieve" instead of "get"

- page 4 line 3: "for the European domain"

- page 5 line 14: "was tested"

- page 5 line 33: "Tiedtke/Bechthold" (with a "t")

- page 6 line 34: "British Isles"

- page 7 line 12: "for the purpose"

- Chapter 4 "Results": Past tense is used for describing the simulation results in this
chapter. Use of present tense would be more appropriate and clearer in my opinion.

- page 8 line 12: ". . . British Isles, Mid-Europe, . . ."

―――――――――――――――

---

## Referee Comment (RC2) · Anonymous Referee #2 · 1 Sep 2020

Dear Editor

I have reviewed the manuscript entitled "ICON in Climate Limited-area Mode (ICON Release Version 2.6.1): a new regional climate model" by Van Pham et al.

The manuscript presents a new regional climate model (RCM) based on the recent ICON model family developed by DWD and other institutes in Germany. This model is thought to replace CCLM. The manuscript recalls the context and history of this new development, presents the model configuration and the simulation setup for an evaluation run and shows the evaluation results with respect to state-of-the art reference datasets (observation-based gridded products) and a CCLM reference run. The birth

of a new RCM is rare and worth to be published. GMD seems to be adapted for such publication. The manuscript is generally well written, well organised and often informative. However I feel that it does not really succeed to full-fill its goals, that is to say to become the foundation of the new RCM for a wide community. In particular, it sometimes looks like a technical note or an internal document of the COSMO-ICON community where not everything is understandable for a reader not participating to it. Also it focuses a lot on the comparison with CCLM for the evaluation run that is not so interesting by itself (a comparison to the Euro-CORDEX RCMs on the whole would be more relevant) whereas at contrary the technical tests of the model are not fully exploited and illustrated in the manuscript.

Finally I'm advising the publication of this manuscript after major revisions. See a detail of my main concerns below as well as a list of minors comments

Main concerns:

1. The manuscript often looks like a technical note of the COSMO community where not everything is understandable for a reader not participating to it. So either this manuscript is a technical note of the COSMO or ICON communities and it does not deserve a GMD publication, or it should be revised to avoid this impression.

1.1 Not all the terms and projects referred within the manuscript are understandable by a reader not used to the COSMO model. Please consider that most of the readers are not part of the COSMO or ICON community and does not know the related projects.

Ex: COPAT project very often mentioned.

Ex: R2B8 configurations and other RxBy.

Ex: Figure 1 is incomplete and not completely self-sufficient. Could you please make the text and the figure 1 consistent: ICON-NWP or ICON-ESM are in the text, not in the figure. Same for the Large-Eddy Simulation version for completeness. Please rework the figure 1.

1.2 Some figures do not have the quality required for a scientific article. It seems that everything was done a bit too quickly without careful final checking by all co-authors.

Ex: figure 2: missing top line. Please defined the PRUDENCE box abbreviations in the captions

Ex: figure 3: names of the variables in brackets are not standard names. Please use standard names and consistent naming between figures, tables and text.

Ex: many figures without units (fig 4, 5, 8, 9, . . .). Please check

Ex: some figures with y-axis labels, other without (cf. Figure 7 vs 8). Please check for consistency.

2. The manuscript focuses a lot on the comparison with CCLM for the evaluation run. I'm not sure that this should be the major point of such an article.

2.1 I understand the will of the authors to focus on this comparison but I would expect that the reader is likely more interested by a fine description of the behaviour of ICLM itself. So at least in the section 4, please spend more time in describing the ICLM behaviour and less time to the description of the CCLM behaviour. You may even want to cut section 4 in two parts, one dedicated to the new model evaluation and the other to the quick comparison with CCLM.

2.2 In addition, personally I would find more relevant to compare ICLM with all the EURO-CORDEX evaluation simulations performed at 12 km. You can either use the results obtained in Kotlarski et al. (2014) if values are numerically available or recompute some of the key scores using data downloaded from the ESGF. Doing so, you will place the newly-developed RCM within the state-of-the-art of the RCMs in Europe. I know that this request requires massive additional work but I hope that the authors will consider it.

Kotlarski, S., Keuler, K., Christensen, O. B., Colette, A., Déqué, M., Gobiet, A., ... & Nikulin, G. (2014). Regional climate modeling on European scales: a joint standard

evaluation of the EURO-CORDEX RCM ensemble. Geoscientific Model Development, 7, 1297-1333.

2.3 Reading the text, I often feel that the authors are "too proud" of ICLM being so close or even better than CCLM. Again I understand the author point of view after so many years of work and the fear of not being as good as the old model. However the way it is phrased is not scientific (objective) enough and show too much satisfaction with themselves. See for example, conclusion, abstract, page 7 line 28-30. Please re-read the whole manuscript and rephrase keeping in mind that the goal of the paper is to present a first version of the model and not to "kill" the old one. Model developers are often not well placed to judge themselves their new model. At the end, it will be up to the readers and then to the ICLM users to decide if the new model better fit their applications. The future will tell us.

3. In a first paper describing a new RCM, I m expecting much more illustrations concerning the technical tests performed with the model before the evaluation run. Many tests are mentioned (time steps, domain decomposition, different computing system) but not really exploited and illustrated. Even if those tests are very appealing to present, for me, they should be at the heart of such paper and each test should be documented by a table or a figure. Currently we need to trust the authors blindly concerning the test results without any proof or trace.

3.1 I'm advising to create a section dedicated to the model tests. That is to say to split section 3.1 in two sub-sections, one describing the tests and one for the evaluation run.

3.2 I'm also advising to add at least the "1+1=2" test. That is to say, checking if running 2 months in one job or in two jobs with a restart between the months give the same results or not. This allows to verify the restarting procedure.

3.3 Later (not for this specific article), I'm also advising to test the model in the Big-Brother / Little-Brother framework what is for me a mandatory step for any new RCM (see for example Denis et al. 2002)

Denis, B., Laprise, R., Caya, D., & Côté, J. (2002). Downscaling ability of one-way nested regional climate models: the Big-Brother Experiment. Climate Dynamics, 18(8), 627-646.

4. Not enough information on the model configuration and simulation setup. In such article, I'm expecting more information about the model itself and its configuration for the evaluation run. The information given in section 2.1 and in section 3.1 are not complete for me.

4.1 First, clarify what should fit in section 2.1 and what should fit in section 3.1. For me everything general concerning the model itself should go in 2.1 whereas the specific model setup for the simulation (domain, resolution, time step, physical choice, tuning, forcing choice) should go in 3.1. The separation is not always easy but deserve some attention to ease the reading.

4.2 For the model description (section 2.1), I'm expecting more information and related tables and figures on the horizontal grid (how does the icosahedric grid look like ?), the distribution of the vertical levels, the output procedure (do you output on the icosahedric grid or on a more classical grid ? See the text page 7, line 11). Do you have the option of spectral nudging in addition to the upper boundary nudging? Also add more information about the relaxation zone and lateral nudging procedure (width, variable nudged, strength of the nudging, filtering tricks if any...) for example in the paragraph page 4 line 12-16.

4.3 For the simulation setup, I'm expecting there the number of grid meshes for the EURO-CORDEX configuration, the way to define the grid, the numerical cost (compared to CCLM at least), the resolution (explain what R2B8 is) but also the description of the forcings of the run. In particular, in addition to the GHG, SST and sea-ice cover (described in section 2.1), I'm expecting some information concerning the aerosol representation (3D+time variation) that can be very variable from one RCM to another (Gutiérrez et al. 2020), the tropospheric ozone and the evolution of the land-use-landcover if any (Davin et al. 2020).

Gutiérrez C., Somot S., Nabat P., Mallet M., Corre L., van Meijgaard E., Perpiñán O., Gaertner M.A. (2020) Future evolution of surface solar radiation and photovoltaic potential in Europe: investigating the role of aerosols. Environ. Res. Lett.,15 (3), 034035, https://doi.org/10.1088/1748-9326/ab6666

Davin, E. L., Rechid, D., Breil, M., Cardoso, R. M., Coppola, E., Hoffmann, P., ... & Raffa, M. (2020). Biogeophysical impacts of forestation in Europe: first results from the LUCAS (Land Use and Climate Across Scales) regional climate model intercomparison. Earth System Dynamics, 11(1), 183-200.

5. A tricky point in RCMs is the capacity to keep or to modify the large-scale information provided by the driving model. Many methods can be applied to check this (Big-Brother/Little-Brother experiment, see above or GCM-RCM temporal or spatio-temporal correlations for large-scale fields often in altitude or cyclone tracking or weather regimes identification). You may want to keep it simple for this study but could you please show at least one illustration allowing to check the lateral forcing procedure? For example, you may want to correlate the Z500 anomaly or the temperature in altitude between the model run and the driver (ERA-Int) at various temporal scales (e.g. yearly, seasonal, monthly, daily, 6-hourly) or anything showing to the reader that ICLM is able to reproduce the large-scale of the driving model at least for some temporal scale (see for example Sanchez-Gomez et al. 2009).

Sanchez-Gomez, E., Somot, S., & Déqué, M. (2009). Ability of an ensemble of regional climate models to reproduce weather regimes over Europe-Atlantic during the period 1961–2000. Climate Dynamics, 33(5), 723-736.

6. Minor comments:

6.1 page 2, line 24: could you explain the difference between "one-way nested sub-domain" and "limited-area mode"? For me, it is the same thing. Is it a question of

on-line versus off-line?

6.2 page 3, line 16: for the update of the SST, could we also use lower frequency such as daily or monthly?

6.2 page 3, line 19: green house → greenhouse

6.3 page 4, line 6-11: this paragraph could perhaps include more information about the input/output procedure, the file format, the flexibility of the outputs, . . . For example, is it possible to output hourly precipitation and monthly-mean MSLP from the same run? or do you need to output all variables at the same frequency before a post-processing step?

6.4 page 4, line 33: grammatical issue

6.5 page 4, line 24-30: could you explain more the restart procedure and the job management and its flexibilit? Could you perform daily run, monthly run, yearly runs ? Or do you have a mandatory time slice such as one month?

6.6 page 5, line 2: could you tell more about the tuning strategy for ICLM. What do you try to optimize?

6.7 page 5 and in many places: EU-CORDEX → EURO-CORDEX

6.8 page 5, line 20: 30 km. Give also the value in hPa.

6.9 page 5, line 29: give the list of the variables nudged and the nudging coefficient

6.10 page 5, line 31-33: The use of many unexplained grid names (R2B8, R3B8, R3B7) is confusing. Simplified or explain. Also in the paragraph, you mention tuning parameters from global settings but setup from LAM. . . clarify

6.11 page 6, line 8: could you compare the 120 s time step with state-of-the-art RCM time steps at the same resolution?

6.12 page 20-23: is the reference CCLM simulation published? Any reference to refer
to? If yes, cite it. If not, you need to describe it in the method section or to use a published run such as one of the EURO-CORDEX evaluation simulations performed with CCLM and available on the ESGF.

6.13 page 6, line 34: clarify that you are considering only land points.

6.14 page 7, line 12: typing issue?

6.15 page 7, line 21: In your case, if I understand well, the RMSE measures a skill related to temporal variations of the variables over the PRUDENCE boxes. So I would have dedicated STDEV to a spatial skill score by averaging in time before computing the standard deviation. Currently STDEV is spatio-temporal score if I understand well, what is therefore quite difficult to interpret. Please, consider to change this. Also table caption mentions "spatial standard deviation" whereas the text mention "spatio-temporal standard deviation". Please clarify.

6.16 page 7, line 21: For the quantitative score, I'm not forcing you to do so but it could have been a better option to plot Taylor diagrams (incl. RMSE, correlation, standard deviation) in order to be more exhaustive in the evaluation of the runs: for example a spatial Taylor diagram per season for all European land points and a temporal Taylor diagram for each PRUDENCE box. This is just an advise. In particular, it allows to put all boxes or all seasons or all variables on the same figure.

6.17 page 7, line 21: If you decide to keep the score STDEV, I propose to put in the tables the ratio of the standard deviations (Model/Obs) in order to have only 2 columns as for the RMSE allowing to easily see the best model for every line.

6.18 page 7, line 28-30: this small paragraph illustrates well my major comment 2.3 with terms such as "very good performance", "consistent for all six evaluated variables", "already of similar". Please rephrase in a more objective and scientific way without overstating the results obtained. Also remember that the ICON project started 20 years ago. So the model is not so new and has been already tuned and adapted at that

resolution over the European domain. I'm aware that a model used in climate mode can show biases not seen in weather forecasting mode but still, you are building on the weather forecast experience. Also note that the model performance is not "consistent" for all variables. From my point of view, it seems better for temperature-related variables than for precipitation or MSLP. Here again, a section comparing the ICON run with all the Euro-CORDEX RCM runs in evaluation mode would be more conclusive (see previous major comment).

6.19 page 8-9-10: Please reorganise the text of those sections to put first the description and discussion of the ICON biases before comparing more quickly with the CCLM reference as the reader want more information about the strengths and weaknesses of ICLM and less about CCLM. Currently I find that the ICLM description is too light and the CCLM description too fat.

6.20 page 8, line 10: "no bias" → When the median bias is near zero, it does not necessarily mean "no bias", it can mean "bias compensation in space". Rephrase.

6.21 page 8, line 17: "extreme daily temperature" → avoid to use the world extreme for min and max daily temperature. It is misleading for the reader as "extreme" is often kept for specific statistics or indices. Check everywhere. Also page 9, line 2, line 3.

6.22 page 8, line 21: "the bias was larger". All the text of the results is written at the past form. I'm not an English specialist but it would be easier to read at the present form → "the bias is larger ...". Please consider to change this everywhere in the results section.

6.23 page 9: please state that ICON is not so good for Summer day statistics. I don't understand why the representation of figure 11 is not similar to the representation of figures 5 to 10 with a box plot representation. A black box can be used for the observation in addition to the green and blue boxes in that case.

6.24 page 9, line 19: not sure I agree that CCLM overestimates the precipitation. It is

relatively well balanced over Europe contrary to ICON.

6.25 page 9, line 20. Please cite a reference for the "too low values". For precipitation, please also mention and discuss the strong model biases over the topography.

6.26 page 9, line 29: "summer had the smallest variations". Not so true if you think that precipitation is very low in summer for some regions. Computing the error in % (even without showing them) may help for discussing the results

6.27 page 9, line 34: "five out of height" → for me it is 7 out of 8. Please check in table 6.

6.28 page 10, line 3-6: for me by eye, CCLM-REF seems better than ICLM-REF for those indices. Please re-assess.

6.29 page 10, line 7: please cut the MSLP and cloud section in two sections, one for each variable for consistency and add the Table 8 for the cloud cover again for consistency.

6.30 page 10, line 8-11: any explanation for the MSLP biases in both models?

6.31 page 10, line 9: for MSLP, it seems that the biases can reach values higher than 2.5 hPa (cf. Figure 4 over Spain for ICLM.

6.32 page 10: same question for the cloud biases. Any explanation or hypothesis?

6.33 page 10, line 20: not clear where you find the +/- 5% values

6.34 page 10, line 20-21: "overestimation of the cloud cover . . . cold bias". ok for the causality for tasmax but this is often the opposite for tasmin. Rephrase.

6.35 page 11, line 11: Personally my assessment is that CCLM is better than ICLM for precipitation. Please re-assess. I agree that models are equivalent for MSLP.

6.36 figure 3: please make this figure easier to read. For example by increasing the thickness of the curves? Possibly showing only seasons or showing maps? Try to

make it simpler and more informative with the key message easier to catch for the reader.

6.37 figure 4: showing the areas where the differences are statistically significant or not may lead to a more informative figure and make it easier to describe in the text in order to focus only on signaificant biases. Please revise the map projection for figure 4 (it is ugly currently) in order to limit the zone without information in each panel. Also "shave your model" that is to say remove the relaxation zone or comment the model behaviour there in the text.

6.38 figure 5-10: I like such figures. Check the y-axis labels and the units everywhere.

6.39 Table 1: Please add more information about the physics by splitting the deep convection and shallow convection lines and by splitting the radiation in short-wave and long-wave radiation. Add in this table all useful information and references for the physics as it will likely serve as reference for many articles afterwards.

6.40 Table 8: please add a table for the cloud cover

---

## Author Response (AR1)

**Dear Referee,**

thank you very much for your time reading our manuscript and the thorough and helpful corrections and recommendations. Thanks to those, we have improved our manuscripts. Here are some of our additional answers to your requests:

- The reader of the paper easily

gets confused with the names of the modelling systems (COSMO, COSMO-CLM, ICON-CLM, ICON-A, ICON-NWP, ICON-LAM, ICON-O, ICON-EUclim, ICON-EU-Nest, ICON-GCM etc). This is in party unavoidable as the names reflect the complex history of the ICON model. Figure 1 is certainly helpful in this respect, but as a reference I would appreciate an additional table that lists the individual configurations of ICON and their basic characteristics. Such a table could also serve as a reference for further publications.

Having a table with individual configurations of ICON is a good idea. However, we would not add an additional table into this paper because it would be a rather complicated table and depart from the main point of introducing and evaluating ICON-CLM here.

- page 1 line 10: "with the setups similar

We changed from setups (plural) to set-up (singular) (page 1, line 11). It fits better in this sentence. In some other parts of the manuscript the plural form setups are still kept after another thought.

- page 2 lines 9-10: It is not really clear to me whether ICON-EU-Nest is a global model with regional refinement over Europe or a higher-resolution version traditionally nested into a global ICON model.

We changed the text. Hopefully it is clearer now (page 2, lines 11-13).

"... ICON-EU-Nest, the regional ICON on European domain interactively nested within the global ICON replaced COSMO-EU (high resolution COSMO model configuration for Europe) for higher-resolution forecasts on the European domain..."

- page 2, line 10: "COSMO-EU" needs an explanation

See previous answer.

- page 2 lines 13-14: As written, the unification was scheduled for end of 2019, which is already in the past. The sentence should hence be modified to "The implies that the last unification of COSMO and COSMO-CLM (COSMO 6), carried out at the end of 2019, was the last one."

The manuscript was written in 2019 but there was a delay in submission till 2020. But the text is also not valid anymore since the plan was changed. The last unification was planned for end of 2020 now. The year was changed accordingly in the text (section 1, page 2, lines 16-17).

- page 2 line 20: Wouldn't it be better to speak of "climate projection" here instead of "climate prediction" to highlight the longest time horizon for application of the model?

Yes "climate projection" was added into the text. "Climate prediction" is still maintained though since they are different (page 2, line 24).

- page 2, line 23: Did ICON-NWP inherit any parts from COSMO (for instance, the physics package)?

The comparison between ICON-CLM and COSMO-CLM is shown in Table 2 and is referred to in Section 3.1. In our opinion, it is not necessary to mention in this part.

- page 2 line 34: Why was CCLM 5.0 clm9 used for the comparison and not the latest(and final) unified version COSMO 6 (see above)?

Because all our work was done in 2018-2019 even before the previous release plan of COSMO 6. The release plan was postponed now till end of 2020. So there is no COSMO 6 yet till this day.

- page 3 lines 13-14: Could you briefly explain why this is the case?

Physically one can feed SST into the regional model also on monthly basis. But we want a flexible option because technically it is easier to update SST at the provided forcing data frequency, so we don't have to prepare specifically the monthly data. The text was reformulated a bit to make things clearer (page 5, line 1-2).

- page 3 lines 20-21: I suggest to replace this expression by "...time dependent GHGsas provided by corresponding GHG scenarios".

Thanks for the suggestion. We changed in the manuscript (page 5, line 9).

- page 5 line 13: It is not really clear which "necessary changes" are meant here.

The text was changed to: "...After the technical adaptation in the ICON model source code to enable long-term climate simulations..." (page 10, line 16).

- page 7 line 12: How were the data transformed/regridded?

We changed the sentence to "... these data were remapped to the lat-lon grids with the same spatial resolution with the observational data ..." (page 9, lines 11-12). Hope it is more understandable now.

- page 7 line 18: Would be better to speak of "ICLM-REF" and "CCLM-REF" here asthe simulations themselves are meant.

Exactly. We changed in the text (page 9, line 20). We also checked again the whole manuscript to make sure that the right names were used.

- page 7 line 28: "a very good performance" -> this statement actually needs somequantification or should, alternatively, be reformulated

We re-structured the manuscript a bit and the sentence was deleted. We also revised all similar statements.

- page 7 lines 28-30: This paragraph actually summarizes the results described later on. It should not precede the presentation of the results I believe, but should rather be moved to the end of the results chapter or even to the conclusions chapter.

The paragraph was moved to the conclusions section. A new small paragraph was written here in this part.

- page 8 line 17: "trends" is misleading here. I'd rather speak of "biases" or "results".

We agreed and changed "trends" to "results" (page 12, line 7).

- page 8 lines 17-18: Any ideas WHY?

It might be due to the positive radiation bias in ICON. We added a discussion into the text (page 12, lines 17-22).

- page 9 line 5: "over the whole evaluation period" -> this expression is misleading as the figure shows no time series of the bias.

Yes, indeed. We changed the text to "averaged over the whole evaluation period" (page 13, line 7).

- page 9 lines 15-16: The numbers obviously refer to events per 20 years. Without providing the length of the period the numbers are, however, not interpretable. I'd suggest to use the unit [days per year] for these numbers and, actually, for the entire Figure 11. This is much easier to understand and to compare to other studies.

Yes that is very true. We changed Figure 11 so that the numbers show days per year instead of per 20 years. The related title and texts were also changed accordingly.

- page 9 line 20: "too low values" -> you're obviously referring to gauge undercatch and evaporative losses here, this should be mentioned (and supported by some reference).

Yes indeed that was what we meant. The text was changed according to your suggestion for better understanding. References for the low precipitation of E-OBS data was also cited (page 13, line 29-32).

- MSLP evaluation in Chapter 4.3: MSLP is evaluated in the same fashion as the other variables, but I see rather little value in this. What is most important here is the spatial MSLP pattern (in addition to the general magnitude), so the evaluation should consider the mean spatial field. The authors might think about replacing their MSLP evaluation by some more informative MSLP analysis.

Thanks for the recommendation. We added an evaluation for MSLP now (Figure 12). The spatial pattern of MSLP from ICLM-REF is compared with the driving data ERA-Interim and additionally with CCLM-REF. A paragraph was also added in the text (page 15 lines 7-17).

- Figure 17: In addition to the naming of the simulations (see above) the variable names in the headers refer to the internal model names of the respective parameters. This is rather cryptic and could be replaced by the full names (2m temperature, cloud coveretc). Alternatively, the abbreviations should be mentioned in the caption.

Perhaps the reviewer referred to Figure 3 (?). We changed the variable names in this figure to full names now. We also revised all naming of the simulations and variables in table, figure, captions, etc. in the manuscript.

- Figure 4: These maps obviously employ some uncommon projection and the Euro-pean continent seems a little "distorted". Why don't you use the same projection as inFigure 2, for instance?

The maps were revised. Hope they look good now.

- Figures 8 and 9: The y-axis of these figures misses a label and the unit of the bias

Sorry it was the result of automatic cropping. All figures should have the full axes now.

- Figure 11: As explained above I'd suggest to use the unit [days per year] or [eventsper year] for these indicators (instead of [days per 20 years]). Furthermore, in topographically structured sub-domains such as AL or SC, the spatial averaging of thenumber of days defined by a temperature threshold makes little sense in my opinion asit completely neglects the large spatial variability. One way to improve on this might beto present the bars as medians with whiskers on top of it reflecting P5 and P95 of thespatial variability within a subdomain. Furthermore, I suggest to place the black EOBSbars to the left of the green simulation bars, not between them (also modify the legendin this case)

Thanks for the comment. We changed the unit from days per 20 years to days per year. However, our main purpose here is to compare the performance of the two models not to look specifically to the spatial variability of the sub-regions. Therefore, we would like to keep the bars as they are. In addition, we also think that adding the whiskers would make the plot very busy and overwhelmed with information. The observation bar we also would like to keep in the middle as it helps with the comparison of both simulations to the reference data.

- Table 1 is hardly readable, the space between the table lines should be enlarged

The space between the table rows were increased, now it is table 2.

**SPELLING AND GRAMMAR**

Thanks for all of the language corrections. We took over most of them. Only one remark below.

- page 1 line 7: "tests"
- page 2 line 1: "the CLM-Community developed"
- page 3 line 16: "with a user-defined"
- page 3 line 19: "the greenhouse gas"
- page 3 line 23: "retrieve" instead of "get"
- page 4 line 3: "for the European domain"
- page 5 line 14: "was tested"
- page 5 line 33: "Tiedtke/Bechthold" (with a "t")
- page 6 line 34: "British Isles"
- page 7 line 12: "for the purpose"

- Chapter 4 "Results": Past tense is used for describing the simulation results in this chapter. Use of present tense would be more appropriate and clearer in my opinion.

Past tense is actually used in scientific writing in Results section, for example Results indicated that ...

Nevertheless, none of the author is native English speaker. We already indicated to the editor that we would agree with a language editing once the manuscript is accepted and in the final form. At this stage there might be more revision, thus it would not bring much to have language editing now.

- page 8 line 12: "...British Isles, Mid-Europe,..."

**Dear Referee,**

thank you very much for your time reading our manuscript and the thorough and helpful corrections and recommendations. Thanks to those, we have improved our manuscripts. Here are some of our additional answers to your requests:

**Main concerns:**

The manuscript often looks like a technical note of the COSMO community where
not everything is understandable for a reader not participating to it. So either this
manuscript is a technical note of the COSMO or ICON communities and it does not
deserve a GMD publication, or it should be revised to avoid this impression.
Thanks to the comments and recommendations, we improved our manuscript tremendously. Hope it looks
better now.

1.1 Not all the terms and projects referred within the manuscript are understandable by a reader not used to the COSMO model. Please consider that most of the readers are not part of the COSMO or ICON community and does not know the related projects. Ex: COPAT project very often mentioned.

We added in the text the explanation of COPAT project (page 8, line 18), we also tried to explain better other terms. Hope it is clearer now.

Ex: R2B8 configurations and other RxBy.

We tried to explain the RxBy better now in the text. A new section "2. General information on ICON-NWP and ICON-LAM" was added to the manuscript to describe the ICON icosahedron grid and the meaning of RxBy. A new table (Table 1) was added to give some description of the grids mentioned in the paper.

Ex: Figure 1 is incomplete and not completely self-sufficient. Could you please make the text and the figure 1 consistent: ICON-NWP or ICON-ESM are in the text, not in the figure. Same for the Large-Eddy Simulation version for completeness. Please rework the figure 1.

True that we are still inconsistent in the naming of the different ICON configurations. We did not intend to include the Large-Eddy configuration and the ICON-ESM into the family tree. But after re-consideration, we revised Figure 1, it should be complete now. And the texts were also changed to give a more complete description (page 2, lines 30-32).

1.2 Some figures do not have the quality required for a scientific article. It seems that everything was done a bit too quickly without careful final checking by all co-authors. Ex: figure 2: missing top line. Please defined the PRUDENCE box abbreviations in the Captions

We changed figure 2 and defined the PRUDENCE box abbreviations in the caption. We also revised all other figures. Please also try to zoom in and out a bit, sometimes the borders of the figures do not show off properly on the pdf viewer.

Ex: figure 3: names of the variables in brackets are not standard names. Please use standard names and consistent naming between figures, tables and text.

We changed the names of the variables in Figure 3 to be consistent with other figures and to be more understandable for the readers.

Ex: many figures without units (fig 4, 5, 8, 9, : : :). Please check

The figures were overcropped. Now they should be find and the units are visible. Ex: some figures with y-axis labels, other without (cf. Figure 7 vs 8). Please check for consistency.

The figures were overcropped. Now the y axis labels are visible.

2. The manuscript focuses a lot on the comparison with CCLM for the evaluation run.

I'm not sure that this should be the major point of such an article.

2.1 I understand the will of the authors to focus on this comparison but I would expect that the reader is likely more interested by a fine description of the behaviour of ICLM itself. So at least in the section 4, please spend more time in describing the ICLM behaviour and less time to the description of the CCLM behaviour. You may even want to cut section 4 in two parts, one dedicated to the new model evaluation and the other to the quick comparison with CCLM.

Thanks a lot for this suggestion, we splitted Section 5 into different sub-sections, one is Technical tests and others for Evaluation and comparison with COSMO-CLM.

2.2 In addition, personally I would find more relevant to compare ICLM with all the EURO-CORDEX evaluation simulations performed at 12 km. You can either use the results obtained in Kotlarski et al. (2014) if values are numerically available or recompute some of the key scores using data downloaded from the ESGF. Doing so, you will place the newly-developed RCM within the state-of-the-art of the RCMs in Europe. I know that this request requires massive additional work but I hope that the authors will consider it.

Kotlarski, S., Keuler, K., Christensen, O. B., Colette, A., Déqué, M., Gobiet, A., … & Nikulin, G. (2014). Regional climate modeling on European scales: a joint standard evaluation of the EURO-CORDEX RCM ensemble. Geoscientific Model Development, 7, 1297-1333.

We agreed with the reviewer that comparing ICLM-REF to the EURO-CORDEX simulations is very tempting. However, we do not consider it a must. Taking this publication below for instance:

GIORGETTA, Marco A., et al. ICON-A, the atmosphere component of the ICON Earth System Model: I. Model description. Journal of Advances in Modeling Earth Systems, 2018, 10. Jg., Nr. 7, S. 1613-1637.

They compared their newly introduced model to a precedent model, in their case ECHAM, and not to many others.

As ICON-CLM is not yet well tuned, it would not be really fair to compare ICLM-REF to the CORDEX ensemble. Nevertheless, we agreed to add a comparison using the figures in Kotlarski et al. (2014). A description was added in the Evaluation methods section (page 10, lines 1-13) and a discussion of the results on page 11, line 28 to page 12 line 6.

We plan a more detailed comparision with other models once the model is fully tuned.

2.3 Reading the text, I often feel that the authors are "too proud" of ICLM being so close or even better than CCLM. Again I understand the author point of view after so many years of work and the fear of not being as good as the old model. However the way it is phrased is not scientific (objective) enough and show too much satisfaction with themselves. See for example, conclusion, abstract, page 7 line 28-30. Please re-read the whole manuscript and rephrase keeping in mind that the goal of the paper is to present a first version of the model and not to "kill" the old one. Model developers are often not well placed to judge themselves their new model. At the end, it will be up to the readers and then to the ICLM users to decide if the new model better fit their applications. The future will tell us.

It has never been the authors' intention to be too proud or to kill the old model. We re-read the manuscript again and still felt that the wording is quite appropriate, perhaps it is just a habit of language. However, we still did some tone down at some places. Hope that it sounds better now.

3. In a first paper describing a new RCM, I m expecting much more illustrations concerning the technical tests performed with the model before the evaluation run. Many tests are mentioned (time steps, domain decomposition, different computing system) but not really exploited and illustrated. Even if those tests are very appealing to present, for me, they should be at the heart of such paper and each test should be documented by a table or a figure. Currently we need to trust the authors blindly concerning the test results without any proof or trace.

The different time step tests were illustrated in Figure 3 and discussed in the text.

We added a description of the domain decomposition test now to the text (page 10 lines 18 to 23). But showing figure or table for these tests is difficult since the results are binary identical. In our opinion, the current text clarifies what we tested and how the results are.

An explanation of the different computing system test was also added to the text (page 11, lines 1-5).

3.1 I'm advising to create a section dedicated to the model tests. That is to say to split section 3.1 in two sub-sections, one describing the tests and one for the evaluation run.
3.2 I'm also advising to add at least the "1+1=2" test. That is to say, checking if running 2 months in one job or in two jobs with a restart between the months give the same results or not. This allows to verify the restarting procedure.

Thanks for this interesting advice. We performed two additional tests according to the referee's recommendation: (1) 2 months in one job without a restart; (2) 2 months in 2 jobs with a restart.

We compared the result of this new tests. The results were identical. A description was added in the text (page 10, line 29-32).

3.3 Later (not for this specific article), I'm also advising to test the model in the Big-Brother / Little-Brother framework what is for me a mandatory step for any new RCM (see for example Denis et al. 2002)

Denis, B., Laprise, R., Caya, D., & Côté, J. (2002). Downscaling ability of one-way nested regional climate models: the Big-Brother Experiment. Climate Dynamics, 18(8), 627-646.

Thanks again for the interesting idea. We will consider that for our next steps with ICON-CLM.

4. Not enough information on the model configuration and simulation setup. In such article, I'm expecting more information about the model itself and its configuration for the evaluation run. The information given in section 2.1 and in section 3.1 are not complete for me.

4.1 First, clarify what should fit in section 2.1 and what should fit in section 3.1. For me everything general concerning the model itself should go in 2.1 whereas the specific model setup for the simulation (domain, resolution, time step, physical choice, tuning, forcing choice) should go in 3.1. The separation is not always easy but deserve some attention to ease the reading.

We looked at section 2.1 and 3.1 again, in our opinion these two sections are already separated, there is no information about the model setup in section 2.1 and vice versa there is no information about the model itself in section 3.1.

4.2 For the model description (section 2.1), I'm expecting more information and related tables and figures on the horizontal grid (how does the icosahedric grid look like ?), the distribution of the vertical levels, the output procedure (do you output on the icosahedric grid or on a more classical grid ? See the text page 7, line 11). Do you have the option of spectral nudging in addition to the upper boundary nudging? Also add more information about the relaxation zone and lateral nudging procedure (width, variable nudged, strength of the nudging, filtering tricks if any: : :) for example in the paragraph page 4 line 12-16.

We added Table 1 to give some charateristics of the grids that were used and referred to in the paper. A description of the icosahadric grid was added into section 2 (page 3, lines 17-27). We also added Figure 2b in addition to Figure 2a to show the triangular grid R2B8. With regards to the vertical level distribution a paragraph was added in section 2 (page 3 line 28 to page 4 line 5). We output on the rotated lat-lon grid not the icosahedric grid, but it can be an option in ICON. This information is added in section 2 (page 4, lines 6-10) and section 4.2 (page 9, line 10).

In ICON limited mode, there are only options for global data nudging or the vertical velocity with the damp layer is damped towards zero. This was already explained in section 2 (page 4, lines 18-22). The information about lateral nudging was added in section 2 (page 4, lines 11-17).

4.3 For the simulation setup, I'm expecting there the number of grid meshes for the EURO-CORDEX configuration, the way to define the grid, the numerical cost (compared to CCLM at least), the resolution (explain what R2B8 is) but also the description of the forcings of the run. In particular, in addition to the GHG, SST and sea-ice cover (described in section 2.1), I'm expecting some information concerning the aerosol representation (3D+time variation) that can be very variable from one RCM to another (Gutiérrez et al. 2020), the tropospheric ozone and the evolution of the land-use-land-cover if any (Davin et al. 2020).

Gutiérrez C., Somot S., Nabat P., Mallet M., Corre L., van Meijgaard E., Perpiñán O., Gaertner M.A. (2020) Future evolution of surface solar radiation and photovoltaic potential in Europe: investigating the role of aerosols. Environ. Res. Lett.,15 (3), 034035, https://doi.org/10.1088/1748-9326/ab6666

Davin, E. L., Rechid, D., Breil, M., Cardoso, R. M., Coppola, E., Hoffmann, P., ... & Raffa, M. (2020). Biogeophysical impacts of forestation in Europe: first results from the LUCAS (Land Use and Climate Across Scales) regional climate model intercomparison.

**Earth System Dynamics, 11(1), 183-200.**

The number of grid meshes for the EURO-CORDEX configuration was added in Table 1. The meaning of R2B8 and general information about the ICON grid was added in section 2, page 3, lines 17-27. The performance in terms of speed with comparison to COSMO-CLM was added in section 5.1, page 11, line 4-5. We have not optimized the run configuration for ICON-CLM, therefore, we would not further comment on the computing cost at this stage. We expect that after an optimization ICON-CLM computational cost would be much less than at the moment.

Information about the aerosol and ozone climatology we used for our simulations was added in section 4.1, page 7, line 14-15.

5. A tricky point in RCMs is the capacity to keep or to modify the large-scale information provided by the driving model. Many methods can be applied to check this (Big-Brother/Little-Brother experiment, see above or GCM-RCM temporal or spatiotemporal correlations for large-scale fields often in altitude or cyclone tracking or weather regimes identification). You may want to keep it simple for this study but could you please show at least one illustration allowing to check the lateral forcing procedure? For example, you may want to correlate the Z500 anomaly or the temperature in altitude between the model run and the driver (ERA-Int) at various temporal scales (e.g. yearly, seasonal, monthly, daily, 6-hourly) or anything showing to the reader that ICLM is able to reproduce the large-scale of the driving model at least for some temporal scale (see for example Sanchez-Gomez et al. 2009).

Sanchez-Gomez, E., Somot, S., & Déqué, M. (2009). Ability of an ensemble of regional climate models to reproduce weather regimes over Europe-Atlantic during the period 1961–2000. Climate Dynamics, 33(5), 723-736.

We calculated the correlation of the geopotential at 500 hPa between ICLM-REF and ERA-Interim data for different time scales (6 hourly, daily, monthly, seasonal, yearly) as suggested. The results are shown in figure 13, a discussion was added in the text as well (section 5.3). In a word, the correlation is pretty high, averaged values higher than 0.925 for all time scales. Correlation is better with longer time scale. ICON-CLM seems to not distort the large scale information of the driving data.

**6. Minor comments:**

6.1 page 2, line 24: could you explain the difference between "one-way nested subdomain" and "limited-area mode"? For me, it is the same thing. Is it a question of on-line versus off-line? Yes correct, that is the difference between "one-way nested subdomain" and "limited-area mode". With "oneway nested subdomain" the nested subdomain and the global domain are being simulated at the same time. Global domain gives forcing to the nested domain, but there is no feedback from nested domain back to global domain.

With "limited-area mode", there is no global model. The boundaries are simply prescribed from external data. We find that the texts are clear enough, and that the referee could already understand the difference, and plus the "one-way nested subdomain" is not the focal point here, we did not change the text.

**6.2 page 3, line 16: for the update of the SST, could we also use lower frequency such as daily or monthly?**

Yes physically one can feed SST into the regional model also on monthly basis. But we want a flexible option because technically it is easier to update SST at the given forcing data frequency, so we don't have to prepare specifically the monthly data. The text was re-formulated a bit to make things clearer (page 5, line 1-2).

**6.2 page 3, line 19: green house ! greenhouse**

We changed from green house to greenhouse in the text.

6.3 page 4, line 6-11: this paragraph could perhaps include more information about the input/output procedure, the file format, the flexibility of the outputs, : : : For example, is it possible to output hourly precipitation and monthly-mean MSLP from the same run? or do you need to output all variables at the same frequency before a post-processing step?

Yes, it is possible to write out different variables with different temporal resolution we added this information in Section 2, page 4, lines 7-10.

6.4 page 4, line 33: grammatical issue

We re-read the line but did not find any grammatical issue. Please re-consider this comment or make it clearer.

6.5 page 4, line 24-30: could you explain more the restart procedure and the job management and its flexibilit? Could you perform daily run, monthly run, yearly runs ? Or do you have a mandatory time slice such as one month?

Yes one can run the model for a wished time period (that is why we could be able to do the 1+1=2 test with two months in one job without a restart). But we normally choose calendar month. Thanks. We added this information to Section 3.2, page 6, lines 17-19.

**6.6 page 5, line 2: could you tell more about the tuning strategy for ICLM. What do you try to optimize?**

What we meant with these text is to introduce the Starter Package of ICON-CLM as an useful tool for different purposes. One is using the Starter Package in tuning ICON-CLM. It is not our intention to tune or to set the strategy of tuning ICON-CLM.

This work is planned in the next phase of COPAT project and will be introduced later.

6.7 page 5 and in many places: EU-CORDEX ! EURO-CORDEX We changed in all places to EURO-CORDEX.

6.8 page 5, line 20: 30 km. Give also the value in hPa. Yes we gave now the value in hPa (page 7, line 11)

6.9 page 5, line 29: give the list of the variables nudged and the nudging coefficient We added this information on Section 4.1, page 7, lines 22-23.

6.10 page 5, line 31-33: The use of many unexplained grid names (R2B8, R3B8, R3B7) is confusing. Simplified or explain. Also in the paragraph, you mention tuning parameters from global settings but setup from LAM: : : clarify
We added the description of the ICON grid in which the names and denotes of the grids are explained in Section 2. Also Table 1 gives information on the mentioned grids.

6.11 page 6, line 8: could you compare the 120 s time step with state-of-the-art RCM time steps at the same resolution?

We do not have information about the time steps from other RCMs, they are also not stated in Kotlarski et al. (2014). Time step of COSMO-CLM at 12 km is 100 s, this information is added on page 8, line 12.

6.12 page 20-23: is the reference CCLM simulation published? Any reference to refer to? If yes, cite it. If not, you need to describe it in the method section or to use a published run such as one of the EURO-CORDEX evaluation simulations performed

with CCLM and available on the ESGF.

The run CCLM-REF is unfortunately not published yet. But it was done with the most recently recommended version and configuration of COSMO-CLM. That's why we chosed this run and not one in EURO-CORDEX evaluation simulations.

The description of the CCLM-REF is in section 4.1, page 7, line 15-23.

**6.13 page 6, line 34: clarify that you are considering only land points. Yes we clarified this in Section 4.2, page 9, line 14.**

6.14 page 7, line 12: typing issue?

Yes indeed. The letter "f" is missing in front of the "or". We corrected now.

6.15 page 7, line 21: In your case, if I understand well, the RMSE measures a skill related to temporal variations of the variables over the PRUDENCE boxes. So I would have dedicated STDEV to a spatial skill score by averaging in time before computing the standard deviation. Currently STDEV is spatio-temporal score if I understand well, what is therefore quite difficult to interpret. Please, consider to change this. Also table caption mentions "spatial standard deviation" whereas the text mention "spatiotemporal

**standard deviation". Please clarify.**

Yes, the sdtdev was spatio-temporal score. Thanks for the advice, we changed it to spatial by calculating the time average and then the deviation. The text was also changed to "spatial". The description of the STDEV was adapted (Section 4.2, page 9, line 29-33). The discussion of the results was also changed accordingly.

6.16 page 7, line 21: For the quantitative score, I'm not forcing you to do so but it could have been a better option to plot Taylor diagrams (incl. RMSE, correlation, standard deviation) in order to be more exhaustive in the evaluation of the runs: for example a spatial Taylor diagram per season for all European land points and a temporal Taylor diagram for each PRUDENCE box. This is just an advise. In particular, it allows to put all boxes or all seasons or all variables on the same figure.

We did actually make the Taylor plots at the beginning, simply because they are part of the Evaluation tool of ICON-CLM and are made automatically when the tool is run. But then we decided to present our results in the form of the tables. Perhaps it is a matter of choice.

6.17 page 7, line 21: If you decide to keep the score STDEV, I propose to put in the tables the ratio of the standard deviations (Model/Obs) in order to have only 2 columns as for the RMSE allowing to easily see the best model for every line. We changed the stdev according to the referee's comment.

6.18 page 7, line 28-30: this small paragraph illustrates well my major comment 2.3 with terms such as "very good performance", "consistent for all six evaluated variables", "already of similar". Please rephrase in a more objective and scientific way without overstating the results obtained. Also remember that the ICON project started 20 years ago. So the model is not so new and has been already tuned and adapted at that resolution over the European domain. I'm aware that a model used in climate mode can show biases not seen in weather forecasting mode but still, you are building on the weather forecast experience. Also not a that the model performance is not "consistent".

weather forecast experience. Also note that the model performance is not "consistent" for all variables. From my point of view, it seems better for temperature-related variables than for precipitation or MSLP. Here again, a section comparing the ICON run with all the Euro-CORDEX RCM runs in evaluation mode would be more conclusive (see previous major comment).

The mentioned small paragraph was removed and replaced.

6.19 page 8-9-10: Please reorganise the text of those sections to put first the description and discussion of the ICON biases before comparing more quickly with the CCLM reference as the reader want more information about the strengths and weaknesses of ICLM and less about CCLM. Currently I find that the ICLM description is too light and the CCLM description too fat.

The manuscript was changed heavily. We hope the ICLM part is better now.

6.20 page 8, line 10: "no bias" ! When the median bias is near zero, it does not necessarily mean "no bias", it can mean "bias compensation in space". Rephrase.In the text, we wrote "nearly no bias" as the medians were about 0.01 K and the percentile boxes were pretty short.

We re-formulated to "relatively small bias" now (page 11, line 21).

6.21 page 8, line 17: "extreme daily temperature" ! avoid to use the world extreme for min and max daily temperature. It is misleading for the reader as "extreme" is often kept for specific statistics or indices. Check everywhere. Also page 9, line 2, line 3. Thanks for the comment. We checked everywhere in the manuscript and replaced the word "extreme" with other expression.

6.22 page 8, line 21: "the bias was larger". All the text of the results is written at the past form. I'm not an English specialist but it would be easier to read at the present form ! "the bias is larger ...". Please consider to change this everywhere in the results section.

The authors are also not English specialists. But from the teaching of tense in academic writting, for results section, past tense is used to describe results obtained. Simple present is used to describe figures, tables. So we would like to keep the tense as it is for the moment.

The manuscript will go through English editting once it is accepted.

6.23 page 9: please state that ICON is not so good for Summer day statistics. I don't understand why the representation of figure 11 is not similar to the representation of figures 5 to 10 with a box plot representation. A black box can be used for the observation in addition to the green and blue boxes in that case.

Yes we added in the text that ICLM-REF is not as good for summer day as for other temperature-related indices. But still it is beter than CCLM-REF in 3/8 sub-regions (ME, SC, AL) and on average over the whole Europe (EU). In three sub-regions, both experiments were equal (BI, IP, FR).

In all figures from 5 to 11, blue is ICLM-REF, green is CCLM-REF, and in figure 11 black is observation. In figure 5 to 10, there is no black because they show already the biases. We could add black boxes representing observation in figures 5 to 11, but since each sub-figure has already 16 boxes, we do not want to make the figures too crowded. We would like to keep figure 5 to 10 as they are, with the biases.

**6.24 page 9, line 19: not sure I agree that CCLM overestimates the precipitation. It is relatively well balanced over Europe contrary to ICON.**

As stated throughout the manuscript, CCLM-REF was better for precipitation. But still it simulated more precipitation in comparison with E-OBS (of course E-OBS properly has measuring error, see the next comment). This overestimation can be seen in figure 4, and even more obvious in figure 11.

**6.25 page 9, line 20. Please cite a reference for the "too low values". For precipitation,**

please also mention and discuss the strong model biases over the topography. We added a citation for the low values due to gauge undercatch and evaporation. The text was also revised a bit too make it clearer what we meant (Section 5.2.2, page 13, lines 30-32).

6.26 page 9, line 29: "summer had the smallest variations". Not so true if you think that precipitation is very low in summer for some regions. Computing the error in % (even without showing them) may help for discussing the results That is true that due to the low precipitation values in summer, the variation of the values are lower than in other seasons. But the statement about the bias was correct.

6.27 page 9, line 34: "five out of height" ! for me it is 7 out of 8. Please check in table

6. The stdev was replaced by the stdev ratio (model/obs). The text was changed accordingly, it is six out of eight.

6.28 page 10, line 3-6: for me by eye, CCLM-REF seems better than ICLM-REF for those indices. Please re-assess.

For Wet days index ICLM-REF was better in 6/9. Heavy precipitation 4/9, very heavy precipitation 3/9. That makes 13/27, and few times the 2 models are give almost the same results. We would keep the same statement that none of the models is better than the other for these indices.

6.29 page 10, line 7: please cut the MSLP and cloud section in two sections, one for each variable for consistency and add the Table 8 for the cloud cover again for consistency.

We cut the MSLP and cloud into two sections (Section 5.2.3, 5.2.4) and table 9 was added for cloud cover.

**6.30 page 10, line 8-11: any explanation for the MSLP biases in both models?**

We do not know. ICLM-REF has a bit higher MSLP than CCLM-REF which shows in its positive MSLP biases, compared to the negative biases from CCLM-REF. The pressure pattern from the driving model is quite well kept in both simulations though.

An additional evaluation for MSLP was added (Figure 12). In this figure, we compared MSLP from ICLM-REF and CCLM-REF to that of the driving model ERA-Interim.

6.31 page 10, line 9: for MSLP, it seems that the biases can reach values higher than 2.5 hPa (cf. Figure 4 over Spain for ICLM.

It is true. The text was changed in page 14, line 28.

**6.32 page 10: same question for the cloud biases. Any explanation or hypothesis?**

We just know from ICON that it produces a bit too little cloud and therefore has a positive bias in radiation. Perhaps this underestimation of cloud is stronger in COSMO-CLM. But we do not know for sure about COSMO-CLM, therefore would not add further comment on this.

**6.33 page 10, line 20: not clear where you find the +/- 5% values**

This line discusses bias of cloud cover in Figure 4. On ICLM-REF side the colors are mostly light pink/blue which correspond to -0.05 to 0.05 and translate to +/- 5%. We added the explanation in the text to make it clearer (Section 5.2.4, page 15, line 21).

6.34 page 10, line 20-21: "overestimation of the cloud cover : : : cold bias". ok for the causality for tasmax but this is often the opposite for tasmin. Rephrase.
What is refered to here is tas. Looking at Figure 4, the overestimation of cloud cover aligns with where CCLM-REF has cold bias for 2 m temperature. We added the reference to Figure 4 to the text to avoid confusion (Section 5.2.4, page 15, line 23).

6.35 page 11, line 11: Personally my assessment is that CCLM is better than ICLM for precipitation. Please re-assess. I agree that models are equivalent for MSLP.It is hard to say from the areal average since some part ICLM-REF is better and some part CCLM-REF is better.But we agree that overall CCLM-REF is better with precipitation and editted the text (Section 6, page 17, line 4)

6.36 figure 3: please make this figure easier to read. For example by increasing the thickness of the curves? Possibly showing only seasons or showing maps? Try to make it simpler and more informative with the key message easier to catch for the reader.

Our idea with this Figure 3 is to test whether ICON-CLM gives remarkably different results due to the choice of time step, which happened with COSMO-CLM. The figure shows that the lines are quite close to the others and no line stands out of the bunch. We would like to keep this figure like that. Of course as written above, we changed the name of the variables and made the sub-figures a bit nicer now.

6.37 figure 4: showing the areas where the differences are statistically significant or not may lead to a more informative figure and make it easier to describe in the text in order to focus only on signaificant biases. Please revise the map projection for figure 4 (it is ugly currently) in order to limit the zone without information in each panel. Also "shave your model" that is to say remove the relaxation zone or comment the model behaviour there in the text.

We revised the map projection and removed the relaxation zone.

6.38 figure 5-10: I like such figures. Check the y-axis labels and the units everywhere. Yes the figures were overcropped. Now y axis labels and units are visible.

6.39 Table 1: Please add more information about the physics by splitting the deep convection and shallow convection lines and by splitting the radiation in short-wave and long-wave radiation. Add in this table all useful information and references for the physics as it will likely serve as reference for many articles afterwards. We added more information regarding shallow/deep convection and short/long wave radiation as requested.

6.40 Table 8: please add a table for the cloud cover

We did. It is table 9.

**ICON in Climate Limited-area Mode (ICON Release Version 2.6.1): a new regional climate model**

Trang Van Pham1, Christian Steger1, Burkhardt Rockel2, Klaus Keuler3, Ingo Kirchner4, Mariano Mertens5, Daniel Rieger1, Günther Zängl1, and Barbara Früh1

1Deutscher Wetterdienst, Frankfurterstr. 135, 63067 Offenbach am Main, Germany
 2Helmholtz-Zentrum Geesthacht, Max-Planck-Str. 1, 21502 Geesthacht, Germany
 3Brandenburg University of Technology, P.O. 10 13 44, 03013 Cottbus, Germany
 4Freie Universität Berlin, Carl-Heinrich-Becker-Weg 6-10, 12165 Berlin, Germany
 5Deutsches Zentrum für Luft- und Raumfahrt, Institut für Physik der Atmosphäre, Oberpfaffenhofen, Germany
 Correspondence: Trang Van Pham (trang.pham-van@dwd.de)

Abstract. For the first time the limited-area mode of the new weather and climate model ICON has been used for a continuous long-term regional climate simulation over Europe. Building upon Built upon the limited-area mode of ICON (ICON-LAM), ICON-CLM (ICON in Climate Limited-area Mode, hereafter ICON-CLM, available in ICON Release Version 2.6.1) is an adaptation for climate applications. A first version of ICON-CLM is now available and has already been integrated into a

- 5 starter package (ICON-CLM\_SP Version Beta1). The starter package provides users with a technical infrastructure that facilitates long-term simulations as well as model evaluation and test routines. ICON-CLM and ICON-CLM\_SP were successfully installed and tested on two different computing systems. Test Tests with different domain decompositions showed bit-identical results, and no systematic outstanding differences were found in the results with different model time steps. ICON-CLM was also able to reproduce the large-scale atmospheric information from the global driving model. Comparison was done between
- 10 ICON-CLM and COSMO-CLM (the recommended model configuration by the CLM-Community) performance. For that, an evaluation run of ICON-CLM with ERA-Interim boundary conditions was carried out with the setups set-up similar to the COSMO-CLM recommended optimal setupsset-up. ICON-CLM results showed biases in the same range as those of COSMO-CLM for all evaluated surface variables. This is remarkable because the While this COSMO-CLM simulation was carried out with the latest model version which has been developed for two decades and was carefully tuned for climate simulations on
- 15 the European domain. Furthermore, ICON-CLM already was not tuned yet. Nevertheless, ICON-CLM showed a better performance for air temperature, its daily extremes, and slightly better for total cloud cover. Results for For precipitation and mean sea level pressuredid not show clear advantage from any model, 
[revised manuscript text omitted]
 2013. Because the outputs of ICLM-REF and CCLM-REF were written on rotated grids that are
- 10 For all ICON-CLM simulations in this paper, the outputs were written in Net-CDF format and on the rotated lat-lon grid as in CCLM-REF. Because this rotated lat-lon grid is finer than E-OBS and CRU grids, these data were transformed to the remapped to the regular lat-lon grids of the observational data or with the same spatial resolution as the observational data for the purpose of comparison. The E-OBS and CRU datasets contain data only over land, therefore the evaluation in this paper (e.g. areal averaged fields) were done using only land points.
- The evaluation within COPAT for 2-m temperature<del>and MSLP, MSLP and cloud cover</del> was done using E-OBS version 10.0 and CRU version 3.22 as reference data. Therefore, in order to compare our evaluation to the one from COPAT, we also used the same versions of the data sets. The comparison period is 20 years from 1981 to 2000, the same as the evaluation period in COPAT. The reference total precipitation data was taken from E-OBS version 12.0 because this dataset (among versions from 10.0 to 17.0) shows the fewest missing data for precipitation over the area of Poland.
- 20 Some important climate indices (listed in Table 3) were calculated from ICON-CLM, COSMO-CLM-ICLM-REF, CCLM-REF and E-OBS 2-m temperature and total precipitation data for the entire and averaged over the period 1981-2000. The number of days that fulfils the definition (in Table 3) was counted for each horizontal grid cell, then averaged over a sub-region.

Root-mean-square error (RMSE) was calculated from the model and observed monthly values (monthly aggregated values for precipitation and monthly mean values for other variables):

25
$$RMSE = \sqrt{\frac{1}{N} \sum_{i=1}^{n} (S_i - O_i)^2}$$
 (3)

where  $S_i$  and  $O_i$  are the model (ICLM-REF or CCLM-REF) and reference data (E-OBS or CRUTS) monthly values, respectively, averaged over the sub-regions considered at the *i*th month; N is the total number of months in the evaluation period 1981-2000.

To compare the spatial variability of the data, spatio-temporal spatial standard deviation (STDEV) was also calculated from 30 monthly time-averaged fields of model and observed datafor all simulation months, and then averaged over time. The STDEV ratio (STDEVmodel/STDEVobservation) was calculated for the ease of comparing the spatial variation of the two model data with respect to the observation. The model with STDEV ratio closer to one better represents the spatial variation of the observation data.

**5 Evaluation and comparison with COSMO-CLM**

In the evaluation run, The results of air temperature from ICLM-REF showed a very good performance . In comparison with reference data , were also compared to those of other regional climate models (RCMs) within the EURO-CORDEX experiments. A thorough evaluation of several EURO-CORDEX ensembles is presented in Kotlarski et al. (2014). One ensemble,

[revised manuscript text omitted]

- 25 of eight sub-regions and weaker variation in four out of eight. In some sub-regions, the STDEV ratio was equal or very close to one (France, Iberian Peninsula, Mediterranean). Compared with CCLM-REFtended to overestimate the spatial variability. This can be most clearly seen in sub-region Eastern Europe with STDEV of 9.03, 8.46 and 8.22 from CCLM-REF, ICLM-REF and E-OBS, respectively. ICLM-REF STDEV were closer to the E-OBS values than CCLM-REF STDEV in all showed in six sub-regions. The differences were especially pronounced in sub-regions Iberian Peninsula and France. better STDEV ratios.
- 30 Similar results were found for minimum 2-m temperature, with ICLM-REF's RMSEs RMSE from ICLM-REF between 0.36 and STDEVs closer to the values of E-OBS 0.85 K. All ICLM-REF's RMSEs were smaller compared to those from CCLM-REF, especially for sub-region Mediterranean and British Isles ICLM-REF errors were less than half (Table 5). ICLM-REF's STDEV ratio was closer to 1 for all sub-regions (Table 5) . in comparison with CCLM-REF. Both models did not simulate well the minimum 2-m temperature spatial variation in mountainous area (Scandinavia and Alps) as for flatter area, but did not
- 35 show a tendency for specific type of orography.

Regarding RMSEs of maximum 2-m temperature (Table 6), in ICLM-REF showed larger biases compared to min/mean 2-m temperature with a maximum bias of 1.54 K in the Mediterranean sub-region. In four out of eight sub-regions, ICLM-REF got lower errors and seven five sub-regions showed better spatial variation. Overall, ICLM-REF simulated average and daily extreme max/min values of 2m air temperature better than CCLM-REF.

- 5 The better-improved representation of daily extreme temperatures max/min 2-m temperature in ICLM-REF resulted in a better performance reduced bias for climate indices that which are determined by air temperature. Among those indices, CCLM-REF overestimated the total annual numbers of ice days and tropical nights over the whole averaged over the evaluation period (1981-2000) as can be seen in Figure 11. Largest tropical night overestimation was found for the sub-region Eastern Europe with 104 nights 5.2 nights per year by CCLM-REF, 6.5 times more than that of E-OBS (16 nights 0.8 nights per year),
- 10 while the ICLM-REF result was much closer to the observations with only 23 nights 1.15 nights per year. Beside that, ICLM-REF results tropical nights indices were very close to the observed numbers for sub-regions France, Alps, Mediterranean, while the numbers in CCLM-REF clearly stand out against the observations. The results for the annual ice days index is similar. were similar; CCLM-REF overestimated the number of ice days for all sub-regions. ICLM-REF, on the other hand, slightly underestimated the annual numbers of ice days but was in all regions sub-regions much closer to the number derived
- 15 from E-OBS than CCLM-REF. The representation of frost days was much better in CCLM-REF compared to the previous two indices, but Generally, ICLM-REF still showed a better performance than showed an underestimation of the annual frost days compared to observations over Europe, except for the Scandinavia sub-region. Compared to CCLM-REFin all eight sub-regions. Both models produced fewer frost days than observed, except for , however, the underestimation was reduced in ICLM-REF. The biggest improvement was simulated in the sub-region Scandinavia. Alps.
- 20 The number of annual summer days was overestimated by ICLM-REF for six of from the eight sub-regions, while CCLM-REF mostly underestimated the amount of summer days. The strongest overestimation of the summer day index compared to CCLM-REF and E-OBS was seen in the Mediterranean sub-region. For three out of eight sub-regions and on average over the whole Europe, ICLM-REF simulated summer day indices more in agreement with observations compared to CCLM-REF. On average, for whole Europe, ICLM-REF resulted in 1065 summer days 53.25 summer days per year; the numbers from E-OBS and CCLM-REF are 1128 and 87456.4 and 43.7, respectively.

**5.3 Precipitation**

**5.2.1 Precipitation**

The mean annual precipitation bias ranged mostly from -50 mm/month to 50 mm/month in both models (Figure 4). Overall, both models simulated more precipitation than E-OBS data. However, one should keep in mind that E-OBS precipitation data

30 tend to give suffer from gauge undercatch and evaporation leading to too low values (Gampe and Ludwig, 2017). According to Kotlarski et al. (2014), the systematic undercatch in E-OBS precipitation data can be on averaged in the order of 4-50 %. That might be the reason why both ICLM-REF and CCLM-REF overestimated precipitation in large part of the domain. CCLM-REF tended to produce too little precipitation than E-OBS along the Atlantic coast while ICLM-REF performed better agreed

better with observation in this area. In the western part of Germany, for example, ICLM-REF had only a slight bias with a difference of less than of 5 mm/month compared with the reference data. CCLM-REF, on the other hand, produced negative biases up to -20 mm/month in this area. However, over the eastern part of Germany, ICLM-REF had larger biases up to more than 10 mm/month. In all other regions, the annual spatial distribution of precipitation biases was quite similar in both models.

- 5 Looking at the spatial variability of the seasonal biases within among the sub-regions in Figure 8, we see that although for some sub-regions in certain seasons, the bias medians were close to zero, the ranges of biases were large. This is expected because precipitation is a highly inhomogeneous variable. Summer and autumn tended to have small median bias in ICLM-REF and CCLM-REF. Among the four seasons, summer had the smallest variation probably due to the low precipitation amount. For winter, summer and autumn, it is not clear which model performed better. For spring, the median and the range of the bias
- 10 were better in better agreement with observations in CCLM-REF compared to ICLM-REF for most of the sub-regions. RMSE for precipitation from ICLM-REF ranged from 10.48 to 30.52 mm (Table 7). The largest error appeared over the Alps sub-region, probably due to the complicated terrain and the dependency of precipitation on orography. RMSEs of CCLM-REF were smaller than those of ICLM-REF for most of the sub-regions, except for sub-region Iberian Peninsula (Table 7). and Scandinavia. ICLM-REF simulated larger variation of precipitation in space compared to E-OBS data, with most STDEV
- 15 ratio larger than 1, except for British Isles with 0.8. The spatial variability of precipitation in CCLM-REF was also closer to observations with better STDEV for five in six out of eight sub-regions indicated by an STDEV closer to one.

ICLM-REF tended to have more precipitation days than CCLM-REF, with the wet days annual wet day index higher for most of the sub-regions and only one exception for sub-region Scandinavia (Figure 11). In six sub-regions, ICLM-REF was more in line with observation than CCLM-REF. ICLM-REF also produced more days with heavy and very heavy precipitation on yearly

20 average than CCLM-REF. For most of the sub-regions, both models overestimated heavy and very heavy precipitation indices, but CCLM-REF was often closer to E-OBS. From our results, it is difficult to judge which model CCLM-REF performed better for precipitation and precipitation related climate indices. The results depended on the area ICLM-REF had in part improvement in certain area of the model domain as well as the season and the index(heavy/very heavy precipitation) considered for certain season and climate index.

**25 5.3 MSLP and cloud cover**

**5.2.1 Mean sea level pressure**

The bias of MSLP of the two models showed opposite signs. ICLM-REF had positive biases while the biases in CCLM-REF were mostly negative as revealed in Figure 4. Although both models had absolute ICLM-REF had biases up to 2.56 hPa, the while CCLM-REF had up to - 4 hPa. The performance of CCLM-REF was betterfor the sub-regions , especially over

[revised manuscript text omitted]
 Radiation short and long wave δ two-stream radiation scheme after RRTM (Rapid Radiative Transfer Model) Land surface and soil scheme TERRA-ML Tiled TERRA Coordinate system horizontal: rotated geographical (lat/lon) vertical: terrain following Gal-Chen height coordinate and exponential height coordinate (SLEVE) according to horizontal:

5 icosahedral grids vertical: terrain following Gal-Chen height coordinate and exponential height coordinate (SLEVE) according to-

Description of climate indices. Index Description Unit Frost days index Number of days with minimum 2-m temperature < 0°C Days Ice days index Number of days with maximum 2-m temperature < 0°C Days Summer days index Number of days with maximum 2-m temperature > 25°C Days Tropical nights index Number of days with minimum 2-m temperature > 20°C

10 Days Wet days index Number of days with total precipitation ≥ 1 mm Days Heavy precipitation days index Number of days with total precipitation > 10 mm Days Very heavy precipitation days index Number of days with total precipitation > 20 mm Days-

RMSE and spatial STDEV ratio of averaged monthly 2-m temperature for the PRUDENCE sub-regions. Data from ICLM-REF, CCLM-REF and E-OBS from 1981 to 2000.

15 Same as Table 4 but for minimum 2-m temperature.
 Same as Table 4 but for maximum 2-m temperature.
 Same as Table 4 but for monthly accumulated total precipitation.
 Same as Table 4 but for MSLP.
 Same as Table 4 but for total cloud cover.

---

## Author Response (AR2)

Dear Referee,

thank you very much for taking the time to review our manuscript and your comments on the revised version.
We have revised our manuscript again and here are some additional answers to your concerns.

The revised version of the manuscript clearly improved on the original one, also thanks to a number of further very useful comments from the second reviewer. The new section on technical tests is indeed very helpful. Also the revised MSLP evaluation is very appropriate. My remaining comments on the original version were satisfactorily accounted for in most cases. Issues that were not worked on by the authors are to some extent a question of taste (for instance, I still believe that a table with the most important characteristics of the individual setups and simulations would be helpful) and it is acceptable to not integrate these suggestions.
A: We already discussed this in the first round among the co-authors. A table like that would be helpful but would be very complicated considering the many configurations and sub-configurations of ICON. We therefore did not add such a table into the manuscript. Thanks for accepting that.

I agree that a final professional language editing would be helpful.
A: Yes we are willing to have language editing once the manuscript is accepted. Indeed we were informed that English language editing is a standard procedure for all manuscripts by GMD. So it would not be a problem.

REMAINING ISSUES

Page 3, line 17: "The resulting grid"
A: We changed the text to „The resulting grid" (page 3, line 17).

Page 3, Eq. 2: Where does this definition originate from and how does the effective grid size relate to the side lengths of the final triangles? Is it much larger? This information would be helpful.
A:
 • We added some more information to show how we come to the definition of the effective grid size (page 3, lines 20-21 and Eq. 2);
 • The effective grid size is indeed around 0.658 size of final triangle side. This information was added to the manuscript together with the fact that this relation can be calculated from Eq. 2. The calculation was not shown though.

Page 3, line 22: I guess it is Figure 2b (not 2c).
A: Thanks for pointing out this mistake. We corrected it now (page 3, line 24).

Page 11, line 22: "is especially large when"
A: We changed according to the referee's suggestion (page 11, line 24).

Page 12, line 27: "can be on average in the order". Also, it might be good not cite Kotlarski et al. in this place but the original works that Kotlarski et al. refer to.
A: We removed the Kotlarski et al. citation and added a citation from Hofstra et al. 2009 „Testing E-OBS European high-resolution gridded data set of daily precipitation and surface temperature" (page 12, line 28).

Page 14, line 1: "a fair comparison"
A: Thanks. We corrected the typing mistake (page 14, line 1).

Page 14, line 6: "was out of the CCM-REF domain"
A: No, the name of the experiment is really CCLM-REF, and an „L".

Page 22, Figure 2b: In the pdf version but especially in any printout the bright lines are very hard to see against the blue background. I suggest to just use a black-and-white version (black lines against white background)
A: Figure 2b is the extraction of the red box area in Figure 2a, it is over the ocean, that's why the background color is blue. We changed the bright lines into black. Hope they look better now!

Page 32, figure legend: "Data were averaged over the"
A: We corrected the typing mistake (page 32, figure 12).